# ST3 beta-galactoside alpha-2,3-sialyltransferase 1 (ST3Gal1) synthesis of Siglec ligands mediates anti-tumour immunity in prostate cancer

Rebecca Garnham[1], Daniel Geh[1], Ryan Nelson[2], Erik Ramon-Gil [1], Laura Wilson[2], Edward N. Schmidt[3,4], Laura Walker [2], Beth Adamson[2], Adriana Buskin[2], Anastasia C. Hepburn[2], Kirsty Hodgson[1], Hannah Kendall[2], Fiona M. Frame[5], Norman Maitland[5], Kelly Coffey [1], Douglas W. Strand[6], Craig N. Robson[2], David J. Elliott[1], Rakesh Heer[2], Matthew Macauley [3,4], Jennifer Munkley[1], Luke Gaughan[2], Jack Leslie [1] & Emma Scott [1] ✉

Immune checkpoint blockade has yet to produce robust anti-cancer responses for prostate cancer. Sialyltransferases have been shown across several solid tumours, including breast, melanoma, colorectal and prostate to promote immune suppression by synthesising sialoglycans, which act as ligands for Siglec receptors. We report that ST3 beta-galactoside alpha-2,3-sialyltransferase 1 (ST3Gal1) levels negatively correlate with androgen signalling in prostate tumours. We demonstrate that ST3Gal1 plays an important role in modulating tumour immune evasion through the synthesises of sialoglycans with the capacity to engage the Siglec-7 and Siglec-9 immunoreceptors preventing immune clearance of cancer cells. Here, we provide evidence of the expression of Siglec-7/9 ligands and their respective immunoreceptors in prostate tumours. These interactions can be modulated by enzalutamide and may maintain immune suppression in enzalutamide treated tumours. We conclude that the activity of ST3Gal1 is critical to prostate cancer anti-tumour immunity and provide rationale for the use of glyco-immune checkpoint targeting therapies in advanced prostate cancer.

Prostate cancer (PC) is the second most common male cancer worldwide, with 1.4 million men diagnosed globally in 2020[1]. During tumorigenesis, prostate tumour growth is driven by androgen receptor (AR) signalling and as such initial therapeutic options for advanced PC are hormone-based therapies, which target AR signalling, such as anti-androgens[2,3]. Most tumours will eventually become resistant to anti-androgen therapies and progress to castrate-resistant prostate cancer (CRPC)[4]. Patients who develop CRPC currently have no curative therapeutic options available to them, and with 375,000 men dying from the disease in 2020 there is a critical need to develop novel therapies for men with advanced PC[1].

An area of innovation in the search for new therapies for CRPC has been immunotherapy. In 2017, Pembrolizumab, an anti-PD-1 agent, was approved for use in solid tumours with high microsatellite instability[5]. Despite this breakthrough, immune checkpoint blockade (ICB) trials have yet to elicit a robust anti-cancer response in PC patients as a monotherapy[6]. There is now a focus on developing new combination therapies, capable of sensitising prostate tumours to ICB, with over 40 clinical trials investigating combination ICB therapies for PC[7–10]. Enzalutamide, a commonly used second-generation AR antagonist can remodel the tumour immune microenvironment (TIME)[11]. Combination

[1]Newcastle University, Centre for Cancer, Newcastle University Biosciences Institute, Newcastle NE1 3BZ, UK. [2]Newcastle University, Centre for Cancer, Newcastle University Translational and Clinical Research Institute, Newcastle NE1 3BZ, UK. [3]Department of Chemistry, University of Alberta, Edmonton, AB T6G 2G2, Canada. [4]Department of Medical Microbiology and Immunology, University of Alberta, Edmonton, AB T6G 2E1, Canada. [5]Cancer Research Unit, Department of Biology, University of York, Heslington, North Yorkshire YO10 5DD, UK. [6]Department of Urology, UT Southwestern Medical Center, Dallas, TX, USA. ✉e-mail: emma.scott@newcastle.ac.uk

enzalutamide-immunotherapies are now in clinical trial for CRPC[12,13]. Results from early phase trials demonstrated a durable therapeutic response in only 18% of participants[12,13]. Clearly, there are underlying mechanisms which prevent cancers responding to current combination therapies. For ICB to be successful in PC, novel treatments need to be developed to target the vast majority of non-responders.

Recently, glyco-immune checkpoints have been identified as drivers of immune suppression in solid tumours and have demonstrated exciting pre-clinical potential as novel targets for combination immunotherapy strategies[14–19]. Siglec receptors are broadly expressed by the immune system and engage with sialic acid to drive immune suppression[20]. Although promising, response to Siglec targeting is dependent upon the local TIME, with immunosuppressive tumours, such as prostate tumours, less sensitive to Siglec targeting[21]. We have previously identified AR-dependent glycosylation changes in PC and have demonstrated that changes in sialylation are a feature of prostate tumours. This includes positive AR regulation of ST6Gal1 and ST6GalNAc1, which were shown to be important for prostate cancer cell survival[22–27]. However, the sialome (all of the sialoglycans in a cell) is highly complex and demonstrates great inter- and intra- patient heterogeneity.

Here, we determine that expression of the sialyltransferase ST3Gal1 negatively correlates with AR signalling in prostate tumours. This led us to investigate the effect of anti-androgen therapies on ST3Gal1 and associated α2-3-linked sialylation patterns. Our results show that in cell models, patient samples and syngeneic mouse models, enzalutamide increases levels of ST3Gal1-driven patterns of α2-3-sialylation. We confirm that ST3Gal1 synthesises immunosuppressive Siglec-7 and Siglec-9 ligands in PC, and that their levels can be modulated by androgen deprivation therapies. Importantly, we identified Siglec-7/9 on immunosuppressive macrophages in prostate tumours and demonstrate that removing their ligands from tumours enhances anti-tumour immunity in a mouse model. We propose that enzalutamide treatment may inadvertently upregulate these suppressive glyco-immune checkpoints and that Siglec targeting therapies may sensitise PC patients to enzalutamide-ICB combination therapies. This highlights the need to understand the cell-type specific glycan-siglec changes in response to systemic therapies for effective disease management.

## Results

### *ST3Gal1* expression inversely correlates with androgen signalling in prostate tumours

Sialylation of core-1 O-glycans has previously been highlighted as a feature of CRPC. Transcriptomic analysis identified ST3Gal1, an enzyme responsible for core-1 O-glycan synthesis, as one of the important glycosyltransferases in CRPC[28]. We sought to investigate the specific role of ST3Gal1 in PC. First, we profiled expression of ST3Gal1 in PC and found protein levels to be significantly increased in prostate tumour tissue compared with healthy normal prostatic tissue (Fig. 1a and Supplementary Fig. 1a). To understand which pathways are altered in prostate tumours with high *ST3GAL1* expression, we performed gene set enrichment (GSEA) on the cancer genome atlas (TCGA) prostate adenocarcinoma (PRAD) cohort[29]. GSEA in 250 patients stratified based on *ST3GAL1* gene expression levels revealed 11 gene sets negatively enriched in *ST3GAL1^high* tumours (Fig. 1b). We noted that the HALLMARK ANDROGEN RESPONSE gene set was negatively enriched in tumours with high expression of *ST3GAL1* (Fig. 1c and Supplementary Table 2).

Given that *ST3GAL1* expression is high in tumours which have low levels of androgen signalling we sought to validate this finding using in vitro models. In the androgen-responsive LNCaP cell line treated with R1881 (an AR ligand) protein levels of ST3Gal1 exhibited a significant decrease compared with steroid-depleted controls (Fig. 1d). In contrast, siRNA knockdown of *AR* resulted in a 3-fold increase in *ST3GAL1* levels (Supplementary Fig. 1b). Several different clinically relevant *AR* isoforms, commonly termed AR variants, have been identified[30–34]. We next looked at the effect of AR variants on the expression of *ST3GAL1*. Selective knockdown of full-length AR and AR variants resulted in a significant increase in

*ST3GAL1* mRNA levels (Fig. 1e). In transcriptomic data from 138 CRPC tumours, levels of *ST3GAL1* gene expression were negatively correlated with *AR, KLK3, NKX3.1* and *TMPRSS2* which are markers of AR signalling activity (Fig. 1f)[35]. This finding was further validated in two independent cohorts in 492 hormone-dependent tumours (Supplementary Fig. 1c) and 208 CRPC tumours (Supplementary Fig. 1d)[36]. AR variants have been linked with the onset of CRPC, and thus we profiled the expression of *ST3GAL1* in CRPC patients. We looked in a publicly available transcriptomic dataset from 59 localised PCs and 35 CRPC patients and found *ST3GAL1* to be significantly higher in CRPC samples (Fig. 1g)[37]. We profiled *ST3GAL1* genomic alterations across four PC cohorts (N=2016) and found *ST3GAL1* is amplified in ~8% of patients in two hormone dependent PC cohorts and amplified in ~20% of patients in two CRPC cohorts (Fig. 1h)[36]. When we stratified 500 patients based on *ST3GAL1* genomic alterations, we found that patients with an *ST3GAL1* amplification have a significantly poorer disease-free survival ($p=0.007$) (Fig. 1i). Taken together, we show that *ST3GAL1* is inversely correlated with AR signalling in prostate tumours and is upregulated in CRPC.

### Androgen receptor antagonism increases ST3Gal1 and α2-3-linked sialoglycans

To further examine the concept that ST3Gal1 is negatively correlated with AR signalling and subsequently increased in CRPC, we asked whether therapeutic targeting of the AR would increase levels of ST3Gal1. LNCaP cells treated with enzalutamide had significantly elevated levels of ST3Gal1, with both mRNA and protein levels increased more than 2-fold (Fig. 2a-b). ST3Gal1 is responsible for the terminal sialylation of core 1 and core 2 O-GalNAc glycans[38]. It catalyses the addition of sialic acid from the nucleotide sugar donor CMP to galactose residues on target glycoproteins through an α2-3-linkage[39]. We quantified cell surface levels of α2-3-linked sialic acid using the Maackia Amurensis Lectin II (MAL-II) lectin, which showed a decrease in α2-3-sialylation in cells treated with neuraminidase (an enzyme which removes sialic acid) (Supplementary Fig. 2b). The flow cytometry gating strategy used is shown in Supplementary Fig. 2a. We observed a significant increase in α2-3 sialylation on the surface of LNCaP cells treated with enzalutamide (Fig. 2c)[40,41].

We next validated our findings in a syngeneic allograft mouse model of PC. The androgen sensitive TRAMP-C2 cell line was implanted subcutaneously in C57BL/6 mice and when tumours were established mice were treated daily with 20 mg/kg enzalutamide for 1 week (Fig. 2d). Enzalutamide treatment resulted in a decrease in tumour growth rate (Fig. 2e) and when excised, tumours were 38% smaller than vehicle treated controls (Fig. 2f). *St3gal1* mRNA levels in enzalutamide treated TRAMP-C2 allografts were significantly upregulated compared with vehicle treated tumours (Fig. 2g). Enzalutamide treatment of TRAMP-C2 cells increased α2-3-sialylation of O-glycans both in vitro (Fig. 2h) and in vivo (Fig. 2i and Supplementary Fig. 2c). Immune phenotyping of vehicle and enzalutamide treated allografts by high-parameter flow cytometry showed re-education of the TIME and we noted a significant 2.14-fold increase in CD8+ T cells (Fig. 2j and Supplementary Fig. 2d). The gating strategy used for immune profiling experiments is shown in Supplementary Fig. 2e. Transcriptomic data on matched patient biopsies treated with enzalutamide showed that *ST3GAL1* mRNA levels were significantly increased post treatment (Fig. 2k)[11]. In the same study, matched paracancerous tissue from patients pre- and post-enzalutamide treatment showed no significant increase in *ST3GAL1* levels following treatment (Supplementary Fig. 2f), suggesting that the observed increase in *ST3GAL1* is specific to prostate tumours. These findings together demonstrate that antiandrogens such as enzalutamide can increase expression of ST3Gal1 and α2-3-sialylation of O-glycans both in vitro and in vivo.

### *St3gal1*-null TRAMP-C2 cells fail to grow C57BL/6 mice

We next determined the effects of St3gal1 on tumour growth in a syngeneic allograft model of PC. We used sgRNAs targeting murine *St3gal1* to

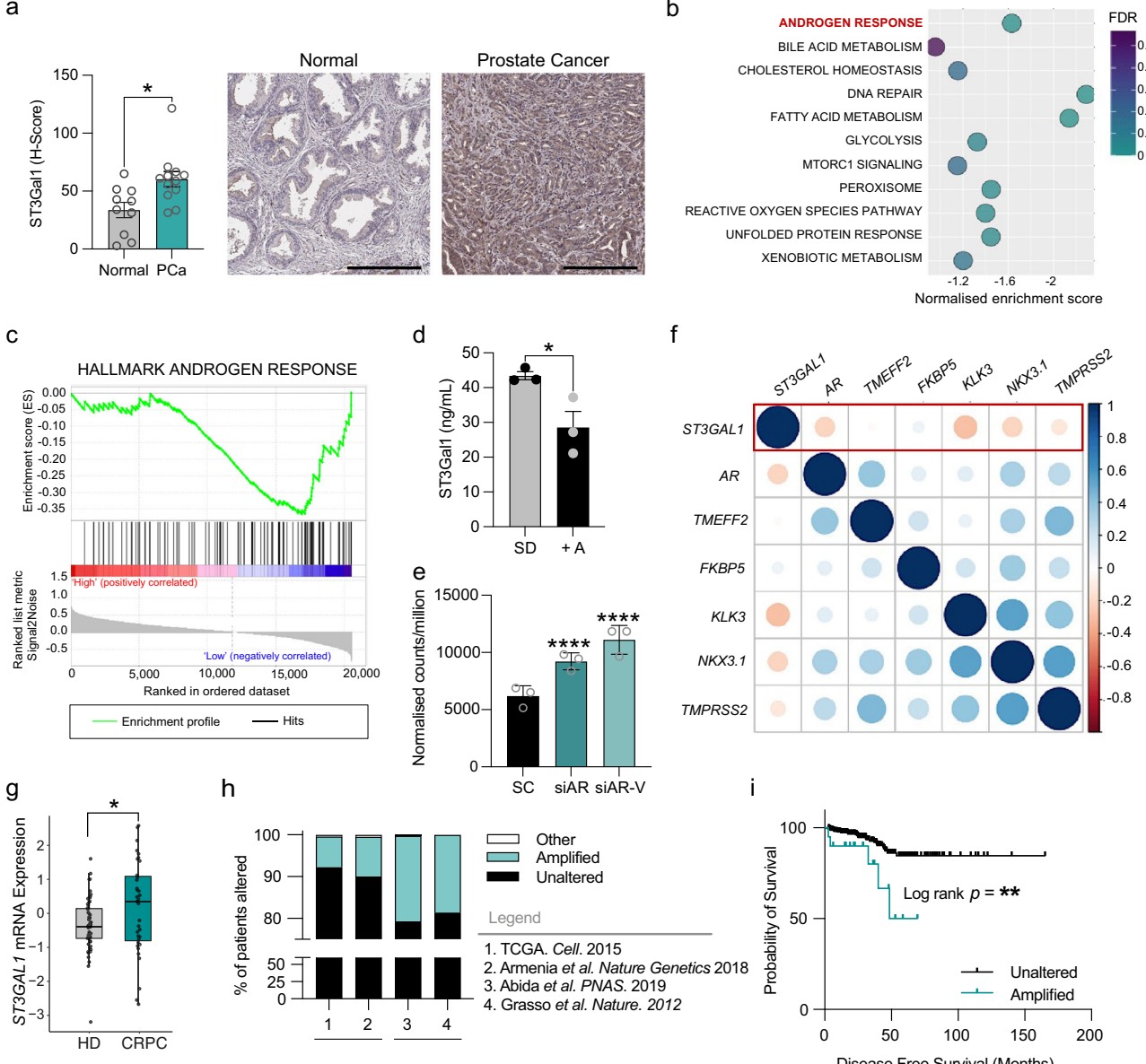

**Fig. 1 | ST3Gal1 expression inversely correlates with androgen signalling in prostate cancer (a)** Immunohistochemical detection of ST3Gal1 protein expression in normal prostate ($N = 10$) and prostate cancer ($N = 12$) tissue samples. Scale bar = 300 μm. **b** Gene set enrichment analysis (GSEA) of The Cancer Genome Atlas (TCGA) Prostate Adenocarcinoma (PRAD) cohort. Patients were stratified based on ST3GAL1 and the top and bottom quartiles compared (N = 250). Pathways negatively enriched in ST3GAL1high patients are shown. FDR = False discovery rate. **c** GSEA for HALLMARK ANDROGEN RESPONSE in TCGA PRAD cohort. **d** Protein level quantification of ST3Gal1 expression in LNCaP cells cultured with or without 10 nm R1881 synthetic androgens (A+) for 24 hours. Protein quantified using a pre-validated ST3Gal1 sandwich ELISA. N = 3 biologically independent samples. **e** Quantification of ST3GAL1 mRNA by RNA sequencing in CWR22Rv1 cells following siRNA knockdown of full-length AR or AR-variants. Statistics shown are adjusted *p* value. N = 3 biologically independent samples.

**f** Correlation matrix correlogram showing ST3GAL1 gene CRPC patients ($N=138$). Pearson's correlation coefficient is shown with −1 (red) to 1 (blue). Only correlations with statistical significance of $p < 0.05$ are shown. The size of the circle is proportional to the correlation coefficients. **g** Normalised ST3GAL1 mRNA levels in publicly available RNA sequencing in patients with CRPC compared to hormone-dependent prostate cancer. **h** Meta-analysis of the percentage of patients with ST3GAL1 genomic alternations across four independent prostate cancer patient cohorts. (TCGA $N = 498$, Armenia et al. N=1013, Abida et al. $N$ 444, Grasso et al. $N=61$). Cohort one and two are representative of hormone-dependent (HD) cancers. Cohorts 3 and 4 represent CRPC patients. **i** Kaplan-Meier plot showing disease-free survival for prostate cancer patients based on unaltered ($N = 314$) and amplified ($N = 20$) ST3GAL1 genomic alterations. Significance tested using: Two-way t-test (**a** and **d**) and Log rank test (**i**). Statistical significance is shown as * $p < 0.05$, ** $p < 0.01$, *** $p < 0.001$ and **** $p < 0.0001$. Error bars show standard error of the mean.

generate *St3gal1−/−* TRAMP-C2 cells. We confirmed successful gene knockout of *St3gal1* in TRAMP-C2 and a subsequent reduction in α2-3-sialylation compared with TRAMP-C2 cells transfected with a non-targeting CRISPR vector (Supplementary Fig. 3a-b). Loss of St3gal1 in TRAMP-C2-C57BL/6 allografts resulted in a 0% engraftment rate, compared with 100% engraftment of non-targeting sgRNA (NT) control cells. When mice containing *St3gal1−/−* TRAMP-C2 cells were culled and

harvested at day 47 there were no signs of early tumour formation (Fig. 3a-b). NT cells grew as expected (Fig. 3c).

We observed no significant difference in cellular proliferation or colony-forming ability between *St3gal1−/−* and NT cells in vitro (Supplementary Fig. 3c-e). Given the role of sialoglycans, and St3gal1 more specifically, in adhesion and integrin biology we assessed whether *St3gal−/−* cells could form three-dimensional (3D) structures in vitro in the form of

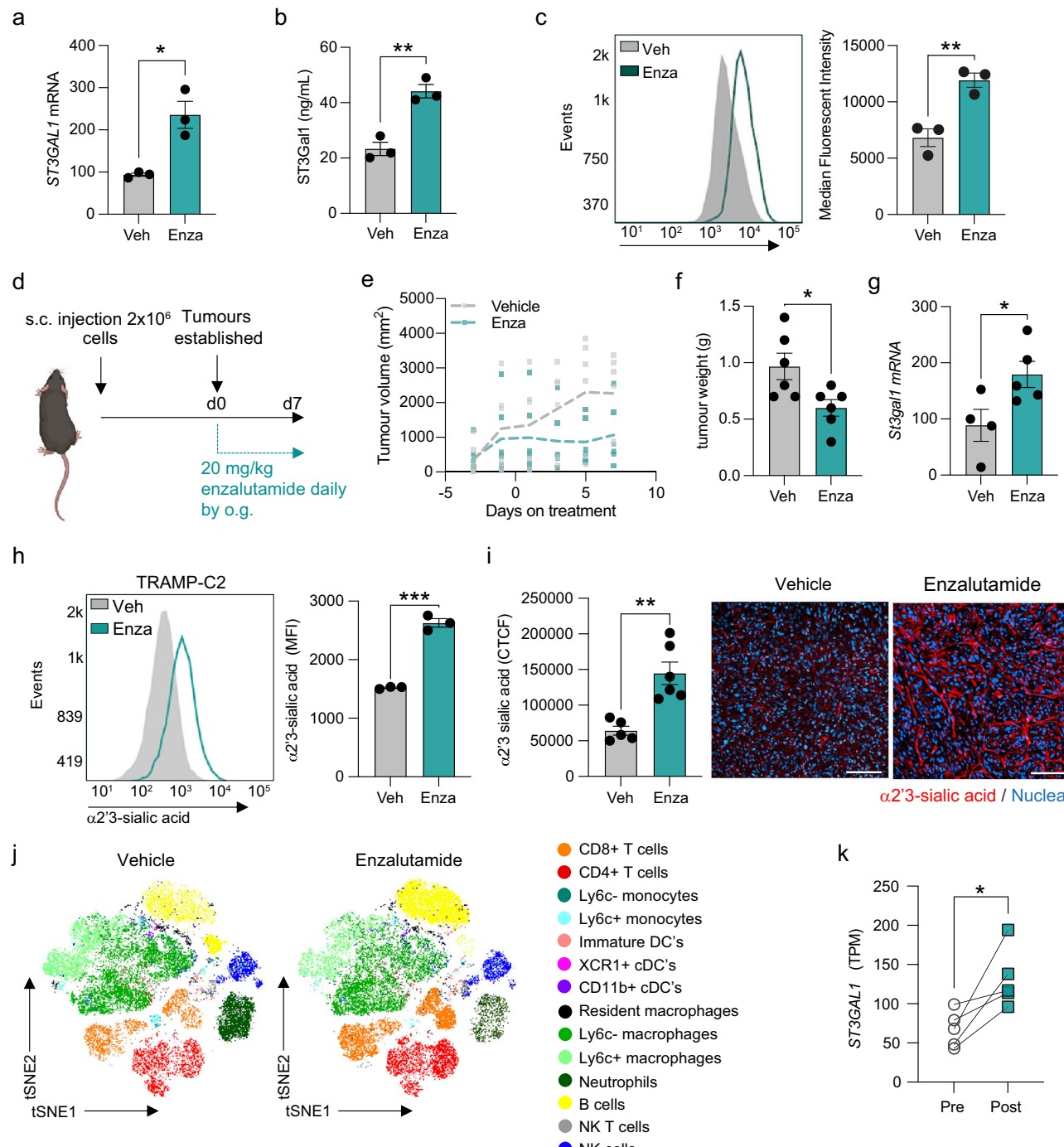

**Fig. 2 | Androgen receptor antagonism increases ST3Gal1 and α2-3-linked sialoglycans (a)** ST3GAL1 mRNA expression in LNCaP cells following 10 μM enzalutamide treatment measured by RT-qPCR. $N = 3$ biologically independent samples. **b** ST3Gal1 protein expression in LNCaP cells following 10 μM enzalutamide treatment quantified using a pre-validated ELISA. $N = 3$ biologically independent samples. **c** MAL-II lectin detection of α2-3-sialylation in LNCaP cells following 10 μM enzalutamide treatment measured by flow cytometry. Representative histogram shown and bar chart of median fluorescent intensities. Histogram representative of $N = 3$ biologically independent samples. **d** Experimental design for TRAMP-C2 subcutaneous allografts in C57BL/6 mice treated with enzalutamide 20 mg/kg daily by oral gavage. Schematic created with BioRender.com **e** Tumour growth curves for subcutaneous allografts with 20 mg/kg enzalutamide treatment or vehicle control ($N = 6$ mice/group). **f** Tumour weights when tumours were harvested following 7 days enzalutamide treatment or vehicle control ($N = 6$ mice/group). **g** RT-qPCR analysis of St3gal1 mRNA expression in TRAMP-C2 subcutaneous tumours following 7 days

vehicle or enzalutamide treatment. **h** MAL-II lectin flow cytometry for cell surface α2-3-sialylation following 10 μM enzalutamide treatment for TRAMP-C2 cells. Representative histogram of $N = 3$ biologically independent samples and bar chart with median fluorescent intensities shown. (**i**) MAL-II lectin immunofluorescence detection of α2-3- linked sialic acid (red) expression in FFPE subcutaneous TRAMP-C2 tumours treated with vehicle or enzalutamide. Data are corrected total cell fluorescence (CTFC). Representative images shown. For each tumour 3 images were taken and quantified. Scale bar = 100 μm. **j** (t-distributed stochastic neighbourhood embedding) tSNE maps of flow cytometric analysis of immune populations in subcutaneous allografts from vehicle and enzalutamide-treated mice ($N = 4$). **k** ST3Gal1 gene expression levels determined by RNA sequencing of match biopsies pre and post enzalutamide treatment ($N = 5$). Significance tested using two-way t-tests. Statistical significance is shown as * $p < 0.05$, ** $p < 0.01$, *** $p < 0.001$ and **** $p < 0.0001$. Error bars show standard error of the mean.

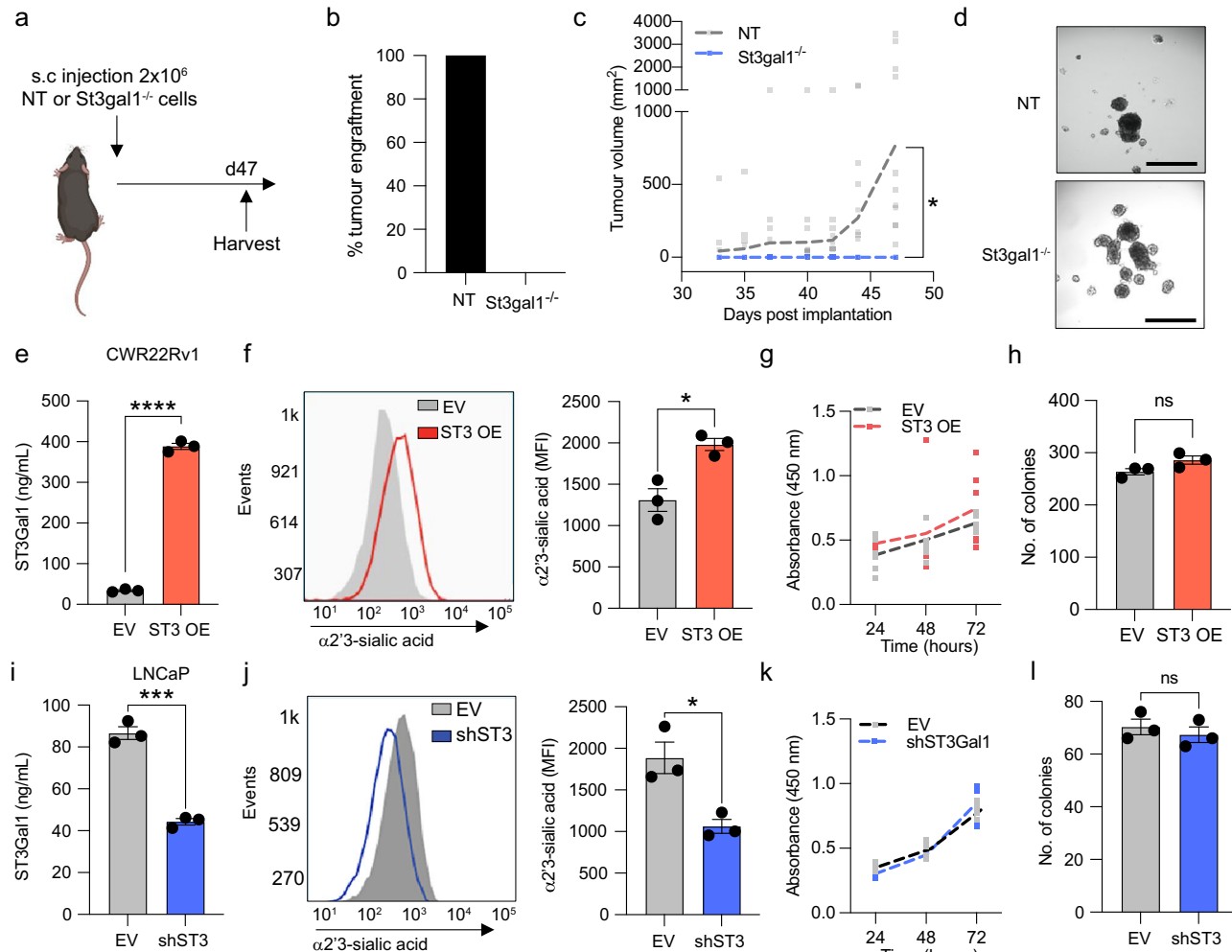

**Fig. 3 | St3gal1-null cells fail to grow in C57BL/6 mice (a) Schematic of St3gal1$^{-/-}$ TRAMP-C2 allograft experimental design.** Schematic created with BioRender.com **b** Percentage tumour engraftment rate for non-targeting (NT) sgRNA control and St3gal1$^{-/-}$ TRAMP-C2 cells. N = 16 mice/group. **c** Tumour growth curves for NT and St3gal1$^{-/-}$ TRAMP-C2 allografts. **d** Representative images of NT and St3gal1$^{-/-}$ TRAMP-C2 spheroid formation in vitro. Images were taken of 9 spheroids per group and quantified. Scale bar = 200 μm. **e** Protein expression of ST3GAL1 in empty vector (EV) and ST3GAL1 overexpression (OE) lentiviral transduced CWR22Rv1 cells. Levels quantified by ELISA. N = 3 biologically independent samples. **f** Quantification of α2-3-sialylation in CWR22Rv1 EV and ST3GAL1 OE cells using the MAL-II by flow cytometry. Representative histogram of N = 3 biologically independent samples and bar chart of median fluorescent intensities. **g** Cellular proliferation of EV and ST3GAL1 overexpression lentiviral transduced CWR22Rv1 cells quantified by WST1 assay. Absorbance was read at 450 nm and normalised to background absorbance. N = 6 biologically independent samples. **h** Colony forming ability of EV and ST3GAL1 overexpression lentiviral transduced CWR22Rv1 cells measured using a colony forming assay. Graph shows the number of colonies formed. N = 3 biologically independent samples. **i** Protein expression of ST3GAL1 in EV and shST3Gal1 knockdown lentiviral transduced LNCaP cells. Levels quantified by ELISA. N = 3 biologically independent samples. **j** Quantification of α2-3-sialylation in LNCaP EV and shST3GAL1 cells using the MAL-II by flow cytometry. Representative histogram of N = 3 biologically independent samples and bar chart of median fluorescent intensities. **k** Cellular proliferation of EV and shST3GAL1 lentiviral transduced LNCaP cells quantified by WST-1 assay. Absorbance was read at 450 nm and normalised to background absorbance. N = 6 biologically independent samples. **l** Colony-forming ability of EV and shST3GAL1 lentiviral transduced LNCaP cells measured using a colony-forming assay. Graph shows number of colonies formed. N = 3 biologically independent samples. Significance tested using two-way t-tests. Statistical significance is shown as * p < 0.05, ** p < 0.01, *** p < 0.001 and **** p < 0.0001. Error bars show standard error of the mean.

spheroids[42–44]. *St3gal1$^{-/-}$* cells did form spheroids and we observed a 26% increase in St3gal1$^{-/-}$ spheroid diameter compared with NT controls, however, this was not significant (Fig. 3d and Supplementary Fig. 3f).

We confirmed our findings in human cell models. We generated stable ST3Gal1 overexpression lines in both CWR22Rv1 and LNCaP cells (LNCaP data shown in Supplementary Fig. 3i-l). We confirmed a significant increase in *ST3GAL1* mRNA (Supplementary Fig. 3g) and ST3Gal1 protein levels (Fig. 3e and Supplementary Fig. 3h). We next demonstrated that ST3Gal1 overexpression resulted in an increase in cell surface α2-3-sialoglycans (Fig. 3f). ST3Gal1 overexpression did not alter cell proliferation or colony-forming efficiency in either CWR22Rv1 (Fig. 3g-h) or LNCaP cells (Supplementary Fig. 3m-p). To confirm our

findings, we generated stable *ST3GAL1* knockdown cells using lentivirus in LNCaP and CWR22Rv1 cells (CWR22Rv1 data shown in Supplementary Fig. 3r-s). We confirmed successful knockdown of *ST3GAL1* at the gene level (Supplementary Fig. 3q), and protein level (Fig. 3i) and a significant reduction in cell surface α2-3-sialylyation (Fig. 3j). In support of our previous findings, gene knockdown of *ST3GAL1* did not affect cellular proliferation or colony forming ability in LNCaP cells (Fig. 3k-l) or CWR22Rv1 cells (Supplementary Fig. 3t-w). When we treated LNCaP cells with ST3Gal1 knockdown with a range of concentrations of enzalutamide we found no significant difference in cell viability when compared with empty vector control cells (Supplementary Fig. 3x). Our findings demonstrate that although we found that ST3Gal1 levels did not

affect proliferative capacity in vitro, St3gal1-null cells fail to grow in immunocompetent mice.

## Siglec-7 and Siglec-9 ligands are synthesised by ST3Gal1 and upregulated by AR targeting therapies

Given our conflicting in vivo and in vitro cell behaviour studies alongside previous studies suggesting that ST3Gal1-associated sialylation promotes tumour immune evasion in breast cancer, we hypothesised that *St3gal1*-null cells failed to engraft in immunocompetent mice as a result of immune clearance[45]. Sialic acid found on tumour cell can act as a ligand for immunosuppressive Siglec receptors[17,20,46–50]. Sialic acid containing glycans capable of engaging Siglec receptors can be probed using Siglec-Fc reagents. We used a panel of commercially available and specifically engineered Siglec-Fcs to profile Siglec ligands in empty vector (EV) and sh*ST3GAL1* knockdown LNCaPs (Fig. 4a and Supplementary Fig. 4a)[51]. For engineered Siglec-Fcs, mutated Siglec-Fcs, that are incapable of binding sialic acid, were used as negative controls (example shown in Supplementary Fig. 4b)[51]. We detected a significant reduction in Siglec-7 and −9 ligands in cells with loss of ST3Gal1. This was independently confirmed in CWR22Rv1 cells (Fig. 4b-c). We also show that overexpression of ST3Gal1 in CWR22Rv1 cells increased surface expression of Siglec-7/9 ligands (Supplementary Fig. 4c). This is in agreement with previous data generated in pancreatic ductal adenocarcinoma cells[18].

Given that ST3Gal1 modulates levels of cell surface Siglec-7 and Siglec-9 ligands in PC cells, we asked whether these ligands are expressed in PC patient biopsies. In prostate tumours, we found that both Siglec-7 and Siglec-9 ligands co-localised with α-methylacyl-CoA racemase (AMACR), suggesting that they are found in cancerous glands within the prostate (Fig. 4d-e and Supplementary Fig. 4d). As our previous data revealed that ST3Gal1 is upregulated by enzalutamide, we hypothesised that anti-androgen therapies would increase levels of Siglec-7/9 ligands. Indeed, surface Siglec-7 and −9 ligand levels increased in LNCaP cells following enzalutamide treatment (Fig. 4f-g). We examined expression of Siglec-9 ligands in fifty patients who were treatment naïve or had received androgen deprivation therapy (ADT). Patients exposed to ADT had a significant 95% increase in expression of immunosuppressive Siglec-9 ligands (Fig. 4h and Supplementary Fig. 4e). We failed to successfully optimise a staining protocol using Siglec-7-FC reagents to detect Siglec-7 ligands in patient tissue. For this reason, it was excluded from our study. We next quantified numbers of Siglec-9+ cells in treatment naïve patients and those who had received hormone therapies and found significantly more Siglec-9+ cells in treated patients (Fig. 4i Supplementary Fig. 4f). We attempted to quantify the number of Siglec-7+ cells in this cohort of patients however attempts to optimise Siglec-7 antibodies for immunohistochemistry on prostate tissue were unsuccessful. We did, however, find that transcript levels of both *SIGLEC7* and *SIGLEC9* were increased in patient post-ADT when compared with matched pre-treatment tissue (Supplementary Fig. 4g). Here we show that ST3Gal1 synthesises key glyco-immune checkpoints in PC, which are upregulated following standard of care anti-androgen therapies.

## ST3Gal1-biosynthesised Siglec ligands are critical glyco-immune checkpoints in prostate cancer

To date, little is known about the expression of Siglec receptors in PC. Siglec-7 and −9 have previously been shown to be expressed on myeloid cells, including macrophages, neutrophils and NK cells[21,49,50,52–54]. Macrophages are the most abundant immune cell type found in prostate tumours and CD163+ macrophages are predictive of a poorer prognosis[55]. Single-cell profiling of prostate tumour-associated macrophages identified 3 distinct populations: pro-inflammatory, anti-inflammatory, and pro-proliferative macrophages with the latter two populations being predictive of a poorer prognosis[56]. In an independent cohort of 208 CRPC patients, we found a strong positive correlation between *SIGLEC7* and *SIGLEC9*, and both markers positively correlated with markers of poorly prognostic macrophages (Fig. 5a). Our analysis identified two clusters of genes which align to the pro-proliferative (highlighted in black box) and anti-inflammatory

(highlighted in red box) macrophage populations and found that both *SIGLEC7* and *SIGLEC9* cluster with an anti-inflammatory macrophage gene signature. In support of our data, re-analysis of single-cell transcriptomic profiling of human prostate tissue confirmed that *SIGLEC7* and *SIGLEC9* is most highly expressed by myeloid cells (Supplementary Fig. 5a-b)[57]. We confirmed that Siglec-9 is co-expressed with CD14+ (a myeloid marker) (Fig. 5b and Supplementary Fig. 5c) and CD163+ (alternatively activated macrophage marker) (Fig. 5c and Supplementary Fig. 5c) in PC patient biopsies. Due to a lack of specific Siglec-7 antibodies that we were confident about, we could not perform co-immunofluorescence experiments for Siglec-7 and myeloid markers.

Transcriptomic analysis of PC patients using camcAPP[58] revealed that mRNA levels of both *SIGLEC7* and *SIGLEC9* are significantly elevated in Gleason grade 9 (4+5) prostate tumours when compared with lower grade tumours (Fig. 5d-e). When we stratified 500 PC patients based on *SIGLEC7* and *SIGLEC9* gene expression, patients with high expression of *SIGLEC7* or *SIGLEC9* had a significantly reduced disease-free survival (Fig. 5f-g). These findings were validated in a second cohort where increased *SIGLEC7* or *SIGLEC9* expression is associated with a reduction in time to biochemical reoccurrence (Supplementary Fig. 5d). Thus, Siglec-7 and −9 are expressed by immunosuppressive macrophages in PC and may contribute to a significantly poorer disease prognosis.

Siglec-E is considered a Siglec-7 and Siglec-9 ortholog/paralog in mice[59]. Siglec-E has been broadly described as a key glyco-immune checkpoint in multiple cancers and targeting of Siglec-E has been shown to repolarise immunosuppressive macrophages towards pro-inflammatory phenotype[15,59,60]. We profiled expression of Siglec-E throughout the TIME in our syngeneic allograft model. We found Siglec-E to be highly expressed by myeloid cells (Fig. 5h). We observed low Siglec-E expression on classic anti-tumour effector cells such CD8+ T cells and NK cells in both the blood and tumour (Fig. 5i). In contrast, we observed high levels of Siglec-E found on intratumoural macrophages. In support of our human data, Ly6C- macrophages, classically thought to be suppressive, exhibited higher Siglec-E expression than Ly6C+ pro-inflammatory subsets.

We next set out to test the hypothesis that ST3Gal1 and its associated sialoglycan patterns interact with the immune system to dampen anti-tumour immunity by selectively depleting key components of the immune system. As the major effector cells of anti-tumour immunity, we targeted CD8+ T cells. We also targeted macrophages given their high expression of the glyco-immune checkpoint Siglec-E. CD8+ T cells and depleted macrophages using anti-CD8α and anti-CSFR1 antibodies, respectively, prior to subcutaneous injection of *St3gal1*-/- TRAMP-C2 cells (Fig. 5j and Supplementary Fig. 5e). As observed previously, *St3gal1*-/- cells implanted in IgG control mice failed to engraft (Fig. 5k). Strikingly, depletion of CD8+ T cells resulted in a 75% engraftment rate of *St3gal1*-/- cells, suggesting that the failure to engraft was, in part, due to CD8+ T cell dependant mechanisms (Fig. 5l). Analysis of tumour growth kinetics showed a delay in tumour growth in anti-CSFR1 treated mice compared with anti-CD8 treated animals (Fig. 5m). However, macrophage depletion also resulted in a 75% engraftment rate, demonstrating a key role for macrophages in mediating *St3gal1* driven immune suppression. Given that macrophages are not conventionally considered to have direct cytotoxic capabilities, we hypothesise that following depletion of St3gal1 and subsequently Siglec-E ligands, macrophages may be re-educated towards an anti-tumour phenotype which could have secondary effects on cytotoxic effector cells such as CD8+ T cells. Finally, we looked at transcriptomic co-expression of *SIGLEC7* and *SIGLEC9* with known immune checkpoints in 208 patients with CRPC (Supplementary Fig. 5f). Transcript levels of both *SIGLEC7* and *SIGLEC9* positively correlate *CD274* (PD-L1), the ligand for PD-1. Of interest, mRNA levels for both immunoreceptors negatively correlates with *CD276* (B7-H3). Further work should look to understand how best Siglec targeting therapies can be combined with a range of conventional immune checkpoint inhibitors to deliver effective immunotherapy approaches for the treatment of advanced prostate cancer.

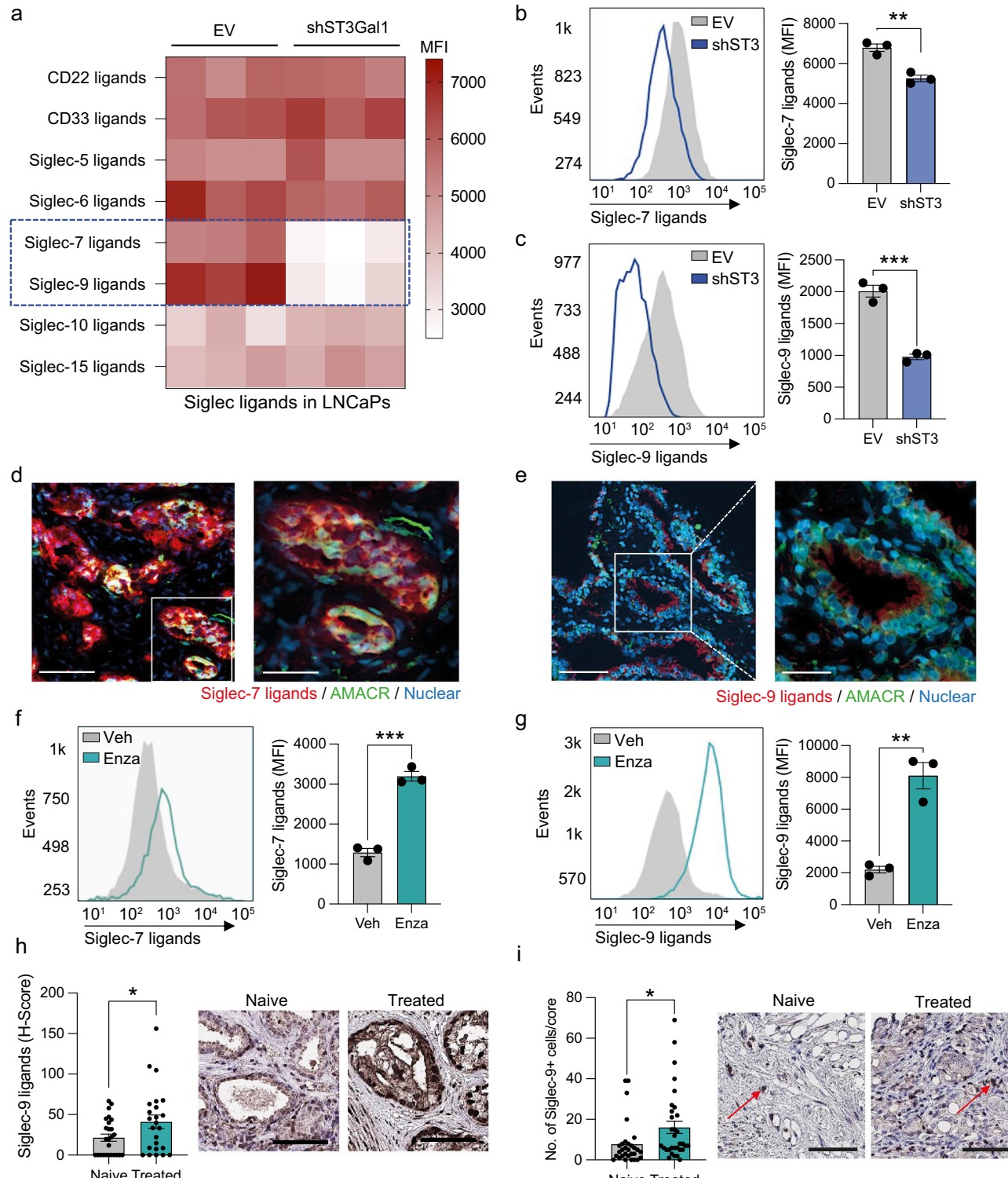

## Discussion

A growing body of literature suggests that targeting glyco-immune checkpoints, specifically Siglec-7/9 may provide therapeutic benefit in several cancers, including Acute Lymphoblastic Leukaemia, pancreatic, breast and melanoma. Therapies which disrupt the sialoglycan-Siglec axis, such as targeted sialidases or Siglec-blocking antibodies have been shown to induce anti-cancer activity in the TIME[15,16,21,50]. However, although there have been multiple studies on glycosylation changes in PC, to date, there is no literature describing the expression of Siglec ligands in prostate tumours or the abundance of tumour associated-Siglec⁺ immune cells. Here, we show that Siglec ligands are expressed in prostate glands and are elevated in patients exposed to hormone-based therapies. We also provide important data on the expression of Siglec-7 and Siglec-9 receptors within the prostate TIME. Critically, we demonstrate that transcript levels of these glyco-immune checkpoints are elevated in aggressive prostate tumours, and high levels of Siglec receptors are associated with a poor disease outcome. Using in vivo models, we have implicated the ST3Gal1-sialoglycan-Siglec axis in macrophage anti-tumour biology and provide proof-of-concept data suggesting

**Fig. 4 | Siglec-7 and Siglec-9 ligands are synthesised by ST3Gal1 and upregulated by AR targeting therapies (a) Heatmap showing siglec binding capabilities in LNCaP empty vector (EV) and shST3GAL1 knockdown cells as determined by flow cytometry using Siglec-Fc reagents.** Significant changes in Siglec-7 and Siglec-9 binding capacity are highlighted in the blue dashed box. N = 3 biologically independent samples. Data are median fluorescent intensities. **b–c** Quantification of Siglec-7 and Siglec-9 binding capacity in CWR22Rv1 EV and shST3GAL1 cells using Siglec-Fc reagents. Representative histogram of $N = 3$ biologically independent samples and bar chart with median fluorescent intensities shown. **d** Siglec-7 ligands (red) colocalized with AMACR (green) in prostate cancer patient biopsies using dual immunofluorescence. Images prepared using a ZEISS Axio Imager2 microscope with a x20 and x40 objective. Scale bar = 150 μm. **e** Siglec-9 ligands (red) colocalized with Alpha-methylacyl-CoA racemase (AMACR) (green) in prostate cancer patient biopsies using dual immunofluorescence. Images prepared using a ZEISS Axio Imager2 microscope with a x20 and x40 objective. Scale bar = 150 μm. **f** Quantification of Siglec-7 ligands using Siglec-Fc reagents in LNCaP cells treated with vehicle or 10 μM enzalutamide. Representative histogram of $N = 3$ biologically

independent samples and bar chart with median fluorescent intensities shown. **g** Quantification of Siglec-9 ligands using Siglec-Fc reagents in LNCaP cells treated with vehicle or 10 μM enzalutamide. Representative histogram of $N = 3$ biologically independent samples and bar chart with median fluorescent intensities shown. **h** Immunohistochemistry detection of Siglec-9 ligands using Siglec-Fc reagents in a tissue microarray (TMA). Patients include those who are treatment naïve ($N = 26$) and those who have been exposed to androgen deprivation therapy ($N = 24$). H-scores were generated to quantify staining in epithelial cells using a Leica Aperio slide scanner. Representative images shown. Scale bar = 300 μm. **i** Immunohistochemistry detection of Siglec-9 in a tissue microarray (TMA). Patients include those who are treatment naïve ($N = 30$) and those who have been exposed to androgen deprivation therapy ($N = 32$). The number of positive Siglec-9$^+$ cells were quantified per tissue core. Representative images shown. Examples of Siglec-9$^+$ ells highlighted with red arrows. Scale bar = 200 μm. Significance tested two-way t-tests. Statistical significance is shown as * $p < 0.05$, ** $p < 0.01$, *** $p < 0.001$ and **** $p < 0.0001$. Error bars show standard error of the mea.

that depleting ST3Gal1 associated sialoglycans or targeting their respective Siglecs may boost immune tumour clearance. These important findings provide the fundamental rationale to study glyco-immune checkpoints as a potential therapeutic strategy for the treatment of advanced PC.

Currently, patients who have advanced PC have no curative options available to them. ICB for the treatment of advanced PC offers some promise, however, to date complete responses to pembrolizumab ICB monotherapy have remained low. Recent reports have demonstrated that enzalutamide treatment has the capacity to reinvigorate the prostate TIME. Some studies suggest that AR targeting therapies may increase numbers of infiltrating immune cells in prostate tumours, although reports on this are conflicting[7,11]. Recent studies trialling pembrolizumab in patients previously treated with enzalutamide (NCT02312557) have highlighted AR activity drives immunosuppression. Crucially, enzalutamide has been shown to act on the AR expressed in T cells to reduce T cell exhaustion, sensitising prostate tumours to ICB[61]. However, this combination is effective in only a minority of patients. Immune suppression for the majority is therefore maintained by AR-independent mechanisms that are yet to be fully elucidated. Importantly, this demonstrates that prostate tumours have the necessary anti-tumour effectors required for an immunotherapy response, they just need to be unlocked.

Hyper-sialylation of solid tumours has previously been shown to be associated with an immunosuppressed TIME. Much of the work studying sialylation of prostate tumours has focused on α2'6-siaylation through the glycosyltransferases ST6Gal1 and ST6GalNAc1, which have been shown to be androgen regulated[22]. In this current study, we show that ST3Gal1 levels negatively correlate with AR signalling in PC however the mechanisms that underpin this remain unclear. Glycosyltransferases are known to be regulated by key oncogenic drivers including the AR. Previous reports on AR regulation of glycosylation have indicated that many of the enzymes involved in glycosylation are positively regulated by androgen signalling[25]. AR can also act as a transcriptional repressor either through recruitment of co-factors or antagonism of other transcriptions factors such as MYC, which has been shown to directly drive ST3GAL1 transcription[62,63]. Reactivation of MYC in response to AR targeting treatments has been identified as a driver of aggressive disease after first line therapy and may be one of the mechanisms which promote a reduction in ST3Gal1 levels when AR signalling is high, and an increase in ST3Gal1 in response to AR therapeutic targeting. Our study therefore demonstrates that AR control of glycosylation is multi-faceted and has therapeutic implications.

Like CD8$^+$ T cells, enzalutamide has been shown to directly affect AR activity in myeloid populations. This, however, drives an immunosuppressive switch, resulting in a pro-tumour macrophage phenotype[64]. In this study, we provide insight into enzalutamide induced immune suppression by showing that enzalutamide treatment increases levels of ST3Gal1 and its associated immunosuppressive Siglec-7 and −9 ligands on the surface of PC cells. Previous studies have shown that Siglec-7 and Siglec-9 ligands are found on O-glycans[60,65–67]. Siglec-7 and −9 and their murine equivalent

Siglec-E have been shown previously to be important glyco-immune checkpoints, directly promoting a protumour macrophage phenotype which can suppress cytotoxic CD8$^+$ T cells[15,21,54]. Taken together these data show that enzalutamide regulation of immune cell phenotype can be both direct, and indirect and can drive pro- and anti-tumour activity in a cell-type specific manner. We propose that this careful balance of pro- and anti-tumour activities can be tilted towards immune-directed tumour clearance by therapies targeting the glyco-immune axis.

In summary, we report that ST3Gal1 synthesises Siglec-7 and Siglec-9 ligands which are critical to maintaining immune suppression in the prostate TIME and that targeting this axis may reactivate anti-tumour immunity. We demonstrate that this important glyco-immune checkpoint is upregulated by AR targeting therapies and may contribute to immune suppression and poor ICB response. Given the complexity of the glyco-immune axis, it will be important to interrogate the effects that system therapies, such as second-generation anti-androgens, have on different cell types including tumour cells, the stroma and individual immune cells. This will be critical to better understand both the therapeutic and unintended off-target effect of current standard-of-care treatments in other cell types. Novel therapies targeting glyco-immune checkpoints are currently being developed and trialled, hence it is timely to determine how PC patients could benefit from these new therapies, and how they may be combined with current standard-of-care treatments.

## Methods
### Human tissue sample ethics
Patient samples were collected with ethical permission from Castle Hill Hospital (Cottingham, Hull) (Ethics Number: 07/H1304/121). Use of patient tissue was approved by the Local Research Ethics Committees. Patients gave informed consent, and all patient samples were anonymized. All ethical regulations relevant to human research participants were followed.

### Bioinformatic analysis of publicly available data
Publicly available transcriptomic datasets were accessed using cBioPortal[68] or camcAPP[58]. Gene set enrichment was performed using GSEA software with available data downloaded from the TCGA PRAD cohort using cBioPortal.

### Cell culture and genetic modification of cell lines
LNCaP and CWR22Rv1 cell lines were routinely cultured in RPMI medium supplemented with 10% foetal calf serum and 1% penicillin-streptomycin. TRAMP-C2 cells were maintained in Dulbecco's modified Eagle's medium with 4 mM L-glutamine adjusted to contain 1.5 g/L sodium bicarbonate and 4.5 g/L glucose supplemented with 0.005 mg/ml bovine insulin and 10 nM dehydroisoandrosterone, 90%; fetal bovine serum (FBS), 5%, Nu-Serum IV, 5%.[24]The stable cell lines used in the study were created by lentiviral

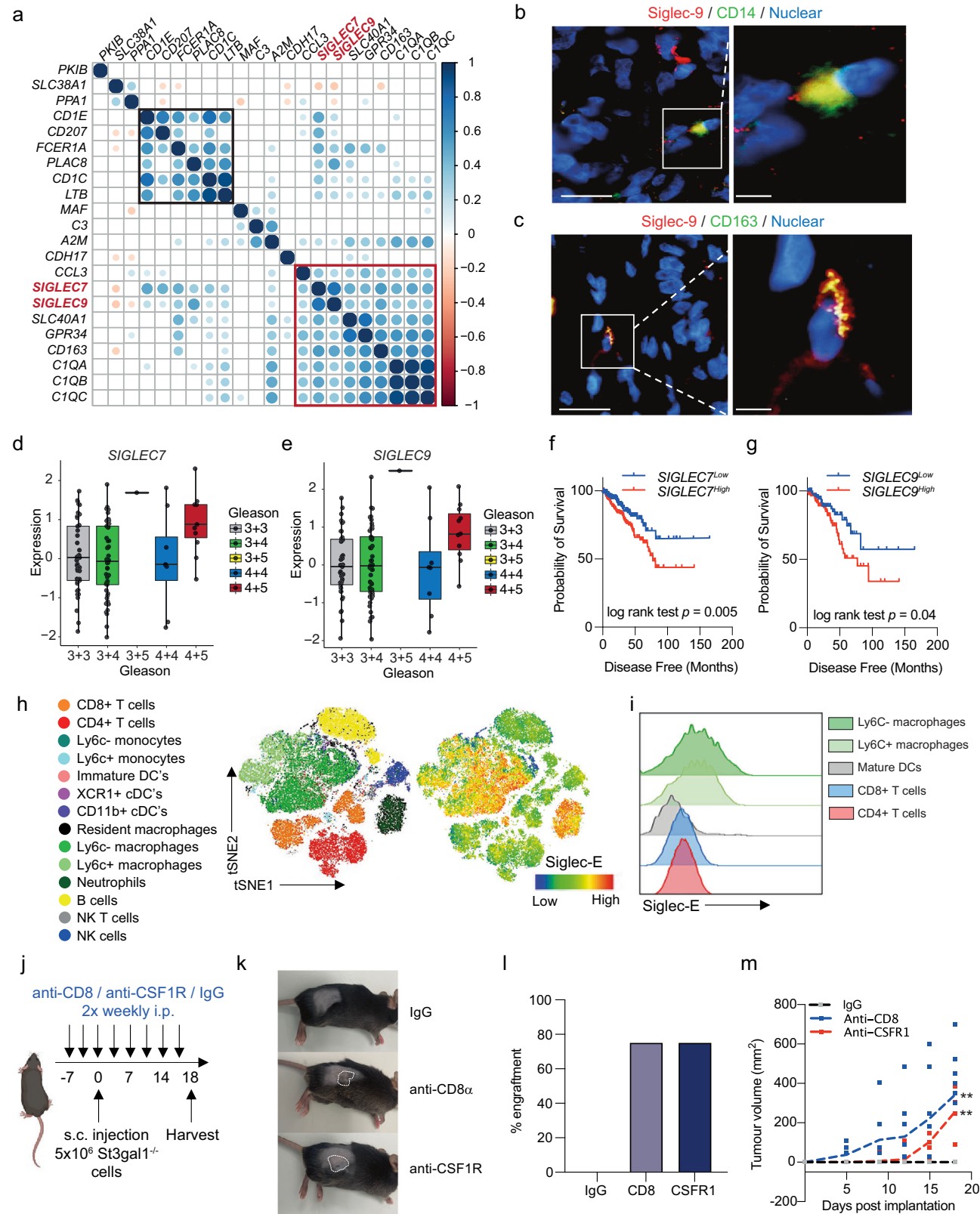

transduction using a multiplicity of infection of 5. For details of the lentiviral particles used see Supplementary Table 3.

For steroid-depleted conditions cells were seeded in RPMI medium with L-glutamine +10% charcoal stripped FBS + 1% Pen/Strep. Cells were treated with 10 μM Enzalutamide (Selleckchem) or 10 nM R1881 for 24 hours.

**Colony formation assay**

$1 \times 10^3$ cells were plated into a single well of a 6-well dish. Colonies were allowed to grow for 3 weeks with the medium replaced at regular intervals. Colonies were fixed with 100% methanol and stained with crystal violet (0.05% w/v). Colony numbers were counted by eye and recorded.

**Fig. 5 | ST3Gal1 bio-synthesised siglec ligands are critical glyco-immune checkpoints in prostate cancer. a** Correlation matrix correlogram correlating mRNA levels of SIGLEC7 and SIGLEC9 with a 19- gene prognostic macrophage signature in 208 CRPC patients in the SU2C dataset. Pearson's correlation coefficient is shown with −1 (red) to 1 (blue). Only correlations with statistical significance of p < 0.05 are shown. Circle size is proportional to the correlation coefficients. A pro-proliferative cluster is highlighted in the black box and anti-inflammatory cluster highlighted in the red box. (**b–c**) Dual immunofluorescence staining of Siglec-9 positive (red) myeloid cells with (**b**) the myeloid marker CD14 (green) and (**c**) alternatively activated macrophage marker CD163 (green) in prostate cancer patient biopsies. Images prepared using a ZEISS Axio Imager2 microscope with a X20 and X63 objective. Scale bars = 20 µm. **d–e** mRNA expression levels of *SIGLEC7* (**d**) and *SIGLEC9* (**e**) from RNA-sequencing of the MSKCC prostate cancer publicly available dataset. Data was accessed through camcAPP. (**f–g**) Kaplan-Meier plot showing disease-free survival for prostate cancer patients stratified based on low (bottom 50%) or high (top 50%) *SIGLEC7* (**f**) and *SIGLEC9*

(**g**) gene expression. Analysis includes 498 prostate cancer patients from the TCGA PRAD cohort, accessed via CBioPortal. **h** t-distributed stochastic neighborhood embedding) tSNE maps of flow cytometric analysis of immune populations in TRAMP-C2 subcutaneous allografts. Siglec-E protein expression on immune cell subsets is shown. **i** Representative stacked histogram of four individual mouse tumours showing Siglec-E expression levels on immune subsets as determined by flow cytometry **j** Schematic of study design for T cell and macrophage depletion studies in St3gal1$^{-/-}$ subcutaneous allografts. Schematic created with BioRender.com **k** Representative photographs taken from mice at the end of the study. Tumours are highlighted with dashed white lines. (**l** Bar chart showing percentage engraftment of St3gal1$^{-/-}$ TRAMP-C2 cells in mice following IgG, anti-CD8 or anti-CSFR1 treatment. **m** Tumour growth curves for St3gal1$^{-/-}$ TRAMP-C2 allografts in IgG control, anti-CD8α and anti-CSFR1 treated mice. Significance tested using: One-way ANOVA (**d,e,l** and **m**) and log rank (**f** and **g**). Statistical significance is shown as * p < 0.05, ** p < 0.01, *** p < 0.001 and **** p < 0.0001. Error bars show standard error of the mean.

## WST-1 assay

5x10$^3$ cells were plated in to a single well of a 96-well culture plate. At 24, 48 or 72 hours WST-1 reagent (Cambridge Bio) was added to each well and incubated at 37 °C for 2 hours. Cell viability was detected at 450 nm wavelength using a Thermofisher Scientific Variskan LUX microplate reader.

## Spheroid experiments

Single-cell suspension was seeded onto the underside of a 15cm culture dish lid, in full growth media, at a density of 3x10$^3$ cells per 20µl. The lid was inverted and placed onto the dish, which contained 10ml PBS. Cells were allowed to form spheroids for four days. Spheroid formation and size was measured using a LeciaDM6 microscope.

## Detection of ST3Gal1 by ELISA

Human ST3Gal1 sandwich pre-validated ELISA kits were purchased from Cambridge Bioscience (RayBioTech, ELH-ST3GAL1). Samples and standards were assayed in duplicate according to the manufacturer's protocol.

## Mouse models

All experiments involving animals received ethical approval and were performed in accordance with a UK Home Office licence (PC02CF4AB), adhered to ARRIVE guidelines and in accordance with the UK Animal (Scientific Procedures) Act 1986. We have complied with all relevant ethical regulations for animal use. All mouse experiments were approved by the Newcastle University Animal Welfare and Ethical Review Board (AWERB). All mice were housed with unrestricted access to food and water and maintained on a constant 12-hour light-dark cycle.

Male C57BL/6 mice (7 weeks old) were purchased from either Envigo or Charles River (UK). For TRAMP-C2 NT and *St3gal1*$^{-/-}$ subcutaneous xenografts, 8-week-old mice were injected subcutaneously with 2x10$^6$ cells in the right flank. For TRAMP-C2 enzalutamide studies, 2 x 10$^6$ TRAMP-C2 cells were engrafted by subcutaneous injection into the right flank of C57BL/6 mice and allowed to establish tumours. Once tumours were established, animals were randomly allocated to vehicle or treatment groups and received a dose of 20 mg/kg enzalutamide or a DMSO vehicle control by oral gavage once daily, at the indicated time point. For immune depletion studies tumour measurements and body weights were taken three times a week. Tumour volume measurements were determined using the formula $l \times w \times h$.

For immune depletion studies mice were randomly allocated to IgG control, anti-CSFR1 or anti-CD8α groups. C57BL/6 mice received 200 µg IgG control, anti-CSFR or anti-CD8 twice weekly by intraperitoneal injection, starting at the indicated time point. 5 x 10$^6$ TRAMP-C2 *St3gal1*$^{-/-}$ cells were injected into the right flank of mice subcutaneously 7 days after depletion began.

## Flow cytometry on tumours and blood

Blood samples were collected into EDTA-coated tubes (BD Biosciences) and treated with lysing buffer (BD Biosciences). Tumours were collected into cold PBS then manually cut into small pieces and digested in Gentle-MACS C dissociation tubes using the GentleMACS tissue dissociator (Miltenyi Biotec) with the manufacture's enzymes (liver dissociation kit, Miltenyi Biotec). Following generation of single-cell suspension enzymes were neutralised with cold RMPI and passed through a 100µm cell strainer. Debris/ dead cells were removed using a 30% percoll gradient and centrifugation. Tumour single-cell suspensions were then treated with RBC lysis buffer (BD Biosciences). Single-cell suspensions were stained with cell viability dye (Invitrogen, LIVE/DEAD fixable blue dead cell stain kit) then blocked with anti-CD16/32 purified antibody at 1:100 for 10 min (Biolegend). Samples were stained with directly conjugated antibodies (see below) for 30 min at 4 °C then fixed in 4% paraformaldehyde. All samples were run on the BD FACSymphony flow cytometer using BD FACSDiva™ software. Data was analysed with FlowJo 10.7.1 software. For high dimensional analysis 10,000 random cells from the CD45$^+$Live leukocyte gate from each sample were down-sampled and concatenated. tSNE maps were generated using the tSNE add on, encompassing all parameters excluding FSC, SSC, Dead and CD45.

All antibodies were purchased from Biolegend apart from; CD11b-BV510, SIRPα-BV711, NK1.1-BV750, Ly6G-BUV395, F4/80-BUV661, CD86-BUV563, CD19-BUV805, SiglecE-BUV615 which were purchased from BD Biosciences. Siglec-E panel: CD4-FITC (1:100, RM4-5), Ly6C-PerCP-Cy5.5 (1:800, HK1.4), PD1-PE (1:100, 29F.1A12), CXCR2-PE/Dazzle 594 (1:100, SA045E1), CD103-Pe-Cy5 (1:100, 2E7), CD3-PE-Cy7 (1:100, 17A2), IA/IE-APC (1:100, M5/114.15.2), SiglecG-R718 (1:100, SH1), CD8α-APC-Cy7 (1:100, 53-6.7), CD197-BV421 (1:100, 4B12), CD11b-BV510 (1:100, M1/70), CD45-BV570 (1:100, 30-F11), CD11c-BV605 (1:100, N418), PDL1-BV650 (1:100, 10F.962), SIRPα-BV711 (1:100, P84), NK1.1-BV711 (1:100, PK136), XCR1-BV785 (1:100, ZET), Ly6G-BUV395 (1:100, 1A8), CD86-BUV563 (1:100, PO3), SiglecE-BUV615 (1:100, 750620), F4/80-BUV661 (T45-2342), CD19-BUV805 (1:100, 1D3). CD8 and CSFR1 depletion confirmation panel: CD4-FITC (1:100, RM4-5), Ly6C-PerCP-Cy5.5 (1:800, HK1.4), CD3-PE-Cy7 (1:100, 17A2), CD45-AF700 (1:100, 30-F11), CD8α-APC-Cy7 (1:100, 53-6.7), CD11b-BV510 (1:100, M1/70), CD115-BV711 (1:100, AFS98), Ly6G-BUV395 (1:100, 1A8), F4/80-BUV661 (T45-2342), CD19-BUV805 (1:100, 1D3).

## Lectin flow cytometry

MAL-II expression was analysed using the MAL-II lectin (Vector Laboratories, B-1305-2) conjugated to streptavidin AlexaFluor 647 (Abcam). 1x10$^5$ cells were pelleted by centrifugation, washed in PBS-T, resuspended in 1X carbo-free blocking solution (Vector Laboratories) and labelled with 2 µg/mL MAL-II lectin for 30 min at 4 °C. Cells were washed and stained with streptavidin Alexa Fluor 647 (Abcam) for 15 min at 4 °C, before repeating

washes and resuspending in PBS. Propidium iodine was used to discriminate between live and dead cells and cells were processed through a FACSymphony flow cytometer. Data was analysed and histograms generated using Red Matter App (v2.01) and plotted using a log scale.

## Siglec-Fc flow cytometry

$1x10^5$ cells were pelleted by centrifugation, washed in cold PBS, resuspended in 1X carbo-free blocking solution (Vector Laboratories) and labelled with Siglec-Fcs purchased from R&D or reagents described previously[51]. Provided reagents were pre-complexed to Strep-tactin-AF647 in the dark for 30 min at 4 °C. Purchased Siglec-Fcs were incubated with cells for 30 min at 4 °C, washed 3 times and then stained with an anti-human Alexa Fluor 647 secondary antibody for 15 min Cells were washed 3 times in PBS, incubated with propidium iodine to discriminate between live and dead cells and processed through a FACSymphony flow cytometer.

## Immunocytochemistry

Cells were washed and fixed using 100% methanol at 4 °C. Next, slides were washed in PBS and blocked with 10% goat serum for 1 hour at RT with gentle rocking. After brief washing with PBS-T, cells were incubated with anti-ST3GAL1 (Invitrogen, PA5-21721, 1:200) antibody diluted in 10% goat serum block overnight. Slides were washed extensively with PBS-T and incubated with an Alexa Fluor 594-goat anti-mouse secondary antibody (Invitrogen). Finally, washes were repeated before counterstaining with Hoechst 33342 (Thermo Fisher Scientific). Images were obtained with fixed exposure times using a ZEISS AxioImager 2 microscope.

## Immunohistochemistry

Heat-mediated antigen retrieval was performed in 10 mM citrate pH 6.0 followed by staining with the appropriate antibody. Antibody dilutions are shown in Supplementary Table 1. Sections were counterstained with haematoxylin. H-Scores were calculated using the Aperio Slide Scanner scoring intensity for only epithelial cells with positive staining.

## Immunofluorescence on FFPE tissue

After dewaxing and rehydration in graded alcohol, human prostate cancer FFPE slides were washed with PBS and heated at 121 °C for 15 min in Tris-EDTA for epitope retrieval. Slides were blocked in 10% goat serum for 1 hour at RT and incubated overnight with the primary antibodies in a blocking solution at 4 °C. The following primary antibodies were used: anti-Siglec-9 (Proteintech,13377-1-AP, 1:200) anti-Siglec-7 (Proteintech, 13939-1-AP, 1:200), anti-CD14 (Proteintech, 60253-1-Ig, 1:500), anti-CD163 (Invitrogen, MA511458, 1:500) anti-AMACR (Proteintech,15918-1-AP, 1:500) overnight at 4 °C. After washing in PBS-T, samples were incubated with goat anti-rabbit Alexa Fluor 647 or donkey anti-mouse Alexa Fluor 488 (Invitrogen) for 60 min at room temperature. Images were obtained using a ZEISS AxioImager 2 microscope.

## Western blotting

Protein was extracted using M-PER mammalian protein extraction reagent (Thermo Fisher Scientific, 78501) according to the manufacturer's instructions. 25 µg of protein was diluted 4:1 with Laemmli buffer boiled at 95 °C for 5 min and separated on a 10% gel at 200V. Gels were transferred onto a nitrocellulose membrane at 300 mA for 1 hour and nitrocellulose membranes were then incubated in 5% skimmed milk in PBS-Tween for 1 hour with rotation. Membranes were incubated with primary for 1 hour at room temperature (RT) or overnight at 4 °C with rotation. Membranes were washed 3 times for 5-10 min in PBS-Tween and the appropriate secondary antibodies conjugated to horseradish peroxidase were added to the membrane for 1 hour at RT with rotation. After 1 hour, membranes were washed 3 times in PBS-Tween for 5–10 min each wash. Nitrocellulose membranes were then incubated with enhanced chemiluminescence (ECL) for 1 min and visualised[69]. An approximate measurement of protein size was assessed using Page Ruler pre-stained protein ladders. For details of the antibodies used please see supplementary table 1.

## Quantitative PCR

Quantitative PCR (qPCR) was performed as previously describe[24]. For details of the primers used please see supplementary table 2.

## Statistics and reproducibility

Statistical analyses were performed using GraphPad Prism 8 (GraphPad Software, Inc., San Diego, CA, USA). Statistical significance is shown as $*$ $p < 0.05$, $**$ $p < 0.01$, $***$ $p < 0.001$ and $****$ $p < 0.0001$. Individual sample sizes are defined in individual figure legends.

## Reporting summary

Further information on research design is available in the Nature Portfolio Reporting Summary linked to this article.

## Data availability

All data presented in this study have been deposited to Figshare (10.6084/m9.figshare.24794589). Source data for Figs. 1b, 2c and 3j are available in Supplementary Data. Uncropped and unedited western blots are available in Supplementary Fig. 6. Sources for all publicly available datasets analysed in this manuscript are available in Supplementary Table 4.

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

## Acknowledgements

We thank Michael Robson, Steve Smith and Melanie Williamson from the Newcastle University Comparative Biology Centre (CBC) for technical support with enzalutamide dosing of animals. We thank CBC staff for the animal husbandry, Newcastle University Bioimaging unit staff for support with fluorescent imaging, and Newcastle University Flow Cytometry Core Facility staff for assistance with flow cytometry. This work was funded by a Prostate Cancer UK Travelling Prize fellowship awarded to ES [TLD-PF19-002]. RG is supported by a William Edmond Harker Foundation Studentship. JL is supported by a JGW Patterson Foundation grant and CRUK program grants [grant numbers C18342/A23390 and DRCRPG-Nov22/100007] and an MRC program grant [grant number MR/R023026/1]. DG is funded by the Newcastle Cancer Research UK (CRUK) Clinical Academic Training Programme. ERG. is funded by the W.E. Harker Foundation. ENS is supported by an Alberta Innovates Graduate fellowship. LW and CNR are supported by Cancer Research UK Experimental Cancer Medicine Centre funding [25160]. KH is supported by Prostate Cancer Research and The Mark Foundation for Cancer Research [6961]. DJE is supported by two Biotechnology and Biological Sciences Research Council grants [BB/S008039/1 and BB/W002019/1]. KC is supported by This work was supported by a Movember funded Prostate Cancer UK Career Development Fellowship [CDF12-006]. AH, AB and RH are supported by a Prostate Cancer Foundation Challenge Award [ PCF (ref. 18:CHAL11)]. DWS is supported by the following grant references (RC2 DK129994, R01 DK115477, R01 DK135535). B.A. is supported by the Ken Bell Bursary and JGW Patterson Foundation [12/21 NU009331]. LW, RN and LG are supported by Prostate Cancer Research [PCR-6955]. MM is supported by NSERC and GlycoNet funding. JM is supported by Prostate Cancer UK through a Research Innovation Award and the Bob Willis Fund [RIA16-ST2-011]. Figures made using Biorender.com.

## Author contributions

RG, ES and DG performed the majority of the laboratory-based work and analyses presented in the manuscript. RN, ERG, LW, ENS, LW, BA, AB, AH, KH, HK, DS performed a portion of the experiments and related analyses. FMF, NJM, KC, CNR, DJE, RH, MSM, LG and JL provided reagents or advice on experimental design and manuscript writing. ES, RG and JL conceived the study and designed experiments. RG and ES wrote the manuscript. All authors read the final manuscript. ES and JM provided funding.

## Competing interests

The authors declare the following competing interest: JM & ES are shareholders of GlycoScoreDx Ltd. All other authors declare no competing interests where relevant.
