## [Peer Review File · Communications Biology]

Reviewers' comments:

Reviewer #1 (Remarks to the Author):

ST3Gal1 synthesis of Siglec ligands mediates anti-tumour immunity in prostate cancer.

Suggestions for the manuscript

1. Abstract: Please elaborate a bit more when mentioning about what has been done earlier- line 25
2. Introduction: Describe in brief when mentioning earlier work in prostate to emphasize- line 68. Paraphrasing last paragraph to make more reader-friendly and highlight the importance/ novelty of the study -80-85
3. Results: Why specifically ST3Gal1 has been chosen for the study among glycosyltransferases responsible for sialylation? Line 90
4. Figure 1D: I would be nice to have blots in the main figure section. The legends could have been more appropriately done. For example, A + could be replaced with the appropriate abbreviation. Figure 1E Axis title does not include log fold change of? (May be ST3Gal mRNA). Please address the abbreviations and titles carefully. This figure could go in the supplementary section, breaks the flow of the data/story.
5. Authors suddenly talk about the AR variants without give the context of why authors felt a need of addressing the need of the question AR variants. As reader, I feel that it doesn't add anything substantial to the story. May be AR variant s figure can be added in the supplementary rather in the main figure.
6. The section, 159-163 - the overall sialylation is reduced, was there any placebo taken for the study?
7. Figure 2B Proteins expression levels using should be included in the main figure. The supplementary blots don't meet the standard quality for inferring the result.
8. Please give bit more detail of the number or ratio when referring slight increase in the spheroid's experiments – line 171 figure 3D.
9. Authors refer to the proliferation assay in figure 3G-H, WST assay is based on metabolic based assay, when addressing any metabolic questions its ideal to check proliferation by cell counts or DNA content. Authors may remove the proliferation figure or rather perform the assay as per the suggestion.
10. Figure 4 authors starts to prolife siglec 9 and 7 but in the figure 4H only siglec 9 has been shown. Immunohistochemistry of siglec 7 is missing. Authors needs to comment why it has not it has not been considered.
11. Figure 4D, In the legends section please do include the full form of abbreviation for AMACR so reader doesn't have to investigate the text.
12. Line 220, please elaborate bit more in detail. Line 222: what is the standard of care hormone therapies are? Please elaborate more in detail.
13. Figure 5 B please include ruler when showing the immunofluorescence/ confocal pictures.
14. Paragraph – 269 -277 can be simplified for clear readability.
15. Discussion: line authors mention about the complexity of the glycol immune axis. How do they plan to address cell specific consequences? Authors shall reflect in detail, how this study is adding to already existing knowledge in respect to sialylation or ST3 Gal1 impact in cancer? authors discuss about the complexity line 347 – 349 does not give clear information nor about the current therapeutic therapies neither the unintended consequences.
16. Discussion could be written better in way emphasizing the importance of the study, what it is contributing to existing research and drawbacks clearly. Addressing question in glycobiology is complex and when trying to establish the correlation with cancer is more complicated. Authors can discuss little bit about the specificity where they suggest that glycol-immune check points can be exploited for the new therapies.

Reviewer #2 (Remarks to the Author):

The present manuscript investigates the role of glyco-immune checkpoints in the progression of advanced prostate cancer (PC) and its resistance to immunotherapy. It draws attention to the unmet therapeutic need for castrate-resistant prostate cancer, with novel targets being sought after for better treatment.

The authors explore the potential of the glyco-immune checkpoint, specifically the sialyltransferase ST3Gal1 and its associated α -2-3-linked sialylation patterns, as a novel target for combination immunotherapy. This builds on the premise that prostate tumors have the necessary anti-tumor effectors for an immunotherapy response but are locked by immune suppression, a majority of which is maintained by AR-independent mechanisms.

The study provides compelling evidence of AR-dependent glycosylation changes in PC. The data supporting an inverse correlation between AR signaling and ST3Gal1 expression in prostate tumors is strong. The authors convincingly show that enzalutamide, an anti-androgen therapy, increases levels of ST3Gal1-driven α -2-3-sialylation and synthesizes immunosuppressive Siglec-7 and Siglec-9 ligands in PC. They propose that enzalutamide may inadvertently upregulate these suppressive glyco-immune checkpoints, thereby creating a compelling rationale for considering Siglec targeting therapies in combination with enzalutamide.

The authors demonstrate the implications of ST3Gal1-sialoglycan-Siglec axis in anti-tumor biology. It shows that depleting ST3Gal1 associated sialoglycans or targeting their respective Sigelects may boost immune tumor clearance. However, the specific mechanisms underlying the correlation between AR signaling and ST3Gal1 expression remain unclear and warrant further investigation.

Moreover, the paper highlights the urgent need to better understand the cell-type specific consequences of systemic therapies for optimizing treatment plans. The suggestion of combining therapies targeting glyco-immune checkpoints with current standard of care treatments is relevant and timely given the ongoing development of such therapies.

While a few additional experiments could further strengthen the manuscript, they are not required and should not delay the publication of the manuscript in its current form in any way.

- Seeing the induction of ST3Gal1 expression by enzalutamide-treatment, it would be interesting to compare the efficacy of enzalutamide-treatment on wt and ST3Gal1 knockout/knockdown cells.
- Similarly, does the upregulation of ST3Gal1 affect the response to immunotherapy in combination with enzalutamide?
- Most pathways identified in Fig. 1B are linked to metabolism. Follow-up of targets beside AR signaling might strengthen the overall role of ST3Gal1 in PRAD.
- A single cell/flow cytometric analysis of the expression pattern of Siglec-7/9 receptors on immune cells in human PC samples would add an additional dimension to support the immunofluorescence stainings.
- Similarly, a single-cell analysis of cells expressing Siglec-7/9 ligands might provide valuable insights.

Overall, the manuscript makes significant contributions to the understanding of glyco-immune checkpoints in prostate cancer and their potential as therapeutic targets. Future work should focus on elucidating the precise mechanisms underlying these observations and conducting clinical trials to test the proposed combination therapies.

Reviewer #3 (Remarks to the Author):

In this manuscript Garnham et al. examine an intriguing inverse correlation between expression levels of androgen signaling pathway members and the sialyltransferase ST3Gal1. The authors convincingly demonstrate that androgen receptor antagonists currently used in the clinic increase both the expression of the ST3Gal1 enzyme and the abundance of its' products, 2,3-sialylic acid structures, on the cell surface. Prior reports have shown that these sialic acid structures can act as Siglec ligands, and the authors expand this knowledge by showing that 2-3-sialylated patterns are Siglec-7/-9 ligands on prostate tumors. Finally, they provide some evidence that these ST3Gal1 upregulated Siglec ligands are benefiting cancer growth through immune inhibitory macrophages.

This manuscript has fascinating implications for the future of prostate cancer therapy. In particular the finding that treatment of prostate tumors with an androgen receptor antagonist currently used in the clinic leads to the upregulation of immune inhibitory Siglec ligands and a more inhibited TIME may help explain why a significant percentage of patients are not responsive to conventional immune therapies. While the exact mechanism of how androgen receptor signaling is linked to ST3Gal1 expression is not elucidated in this work, the findings are novel and interesting to the field of biology. The data is well-presented, and a major strength that the authors make the effort to repeat their findings in multiple of cell lines, spheroid models, animal models, as well as using human data and tissue biopsies throughout the manuscript.

I have two requests for further information or possible experiments:

1. In lines 235-240 / Fig. 5B & C. It is mentioned that both SIGLEC7 and SIGLEC9 cluster with an anti-inflammatory macrophage gene signature. The manuscript then confirms Siglec-9 expression colocalization on CD14+ and CD163+ myeloid cells in patient biopsies, but no mention is made of having looked for Siglec-7 expression there. Since the bulk of the paper focuses on both of these receptors it would be ideal to include these results either in the main figure or the supplementary Figs. Alternatively, an explanation of why the authors elected to proceed with only Siglec-9 colocalization could be provided.

2. Similarly, in Figure 4. The bulk of the figure focuses on looking at Siglec-7/-9 ligands on multiple cell lines and in patient biopsies, but for the tissue microarray quantification (Fig. 4H & I) only Siglec-9 is investigated. Ideally, there would also be data for Siglec-7 here or in supplementary data, even if the data are negative, but there may be reasons why this could not be/was not done.

Other minor revisions:

3. In the Figure 1D legend – state how the protein was quantified

4. In Figure 1I legend – could you state how many were in each group (amplified) and (unaltered)

5. In Figures 1A , 2I, 3D, 4D, 4E, 4H, 4I, 5B, 5C, there is either no scale bar, or the scale bar is present but not labeled with a number on the figure or in the legend.

6. For Fig. 2H, this is likely a glitch with this particular upload – but the background for this subfigure is neon yellow in the version uploaded and certain cell populations can't be seen.

7. Figure 2H (right half): Coloring scheme is undifferentiable to red/green colorblind folks, would be preferable to change this.

8. Supplementary Figure 1D has a textbox pasted over the figure.

9. There are some single representative tissue samples/TMA/confocal images in the paper, can some other images collected for these experiments be added to the supplementary figures or to a database ?

10. The manuscript occasionally mentions a representative image in a figure – each time a

representative image is used please include how many replicates/images were taken for each experiment (for example, how many spheroids were imaged for each group in Fig. 3D?)

11. In Figure 3D is there some way to quantify the spheroid sizes in your images?

12. In all histogram-containing figures, I want to verify that the x-axis spacing is correct. If so, what scaling is this? It looks neither log scale nor bi-exponential - it might be good to add the scale information into the methods section or figure legends.

Response to reviewers' comments

Reviewer #1 (Remarks to the Author):

1. Abstract: Please elaborate a bit more when mentioning about what has been done earlier- line 25.

Reply: *We have now changed the text to read: 'Sialyltransferases have been shown across several solid tumours, including breast, melanoma, colorectal and prostate to promote immune suppression by synthesizing sialoglycans, which act as ligands for Siglec receptors.'*

2. Introduction: Describe in brief when mentioning earlier work in prostate to emphasize- line 68. Paraphrasing last paragraph to make more reader-friendly and highlight the importance/ novelty of the study -80-85.

Reply: *We have now added an extra sentence on our previous work in prostate cancer, "This includes positive AR regulation of ST6Gal1 and ST6GalNAc1 which were shown to be important for prostate cancer cell survival". We have also re-written the final paragraph to make it more reader friendly.*

3. Why specifically ST3Gal1 has been chosen for the study among glycosyltransferases responsible for sialylation? Line 90.

Reply: *To help with clarity, we have added the following sentence as tracked changes. "Transcriptomic analysis identified ST3Gal1, an enzyme responsible for core-1 O-glycan synthesis, as one of the important glycosyltransferases in CRPC²⁶. We sought to investigate the specific role of ST3Gal1 in PC."*

4. Figure 1D: I would be nice to have blots in the main figure section. The legends could have been more appropriately done. For example, A + could be replaced with the appropriate abbreviation. Figure 1E Axis title does not include log fold change of? (May be ST3Gal mRNA). Please address the abbreviations and titles carefully. This figure could go in the supplementary section, breaks the flow of the data/story.

Reply: *We confirmed a reduction in ST3GAL1 mRNA and protein using RT-qPCR and ELISA. In this study our primary focus was quantifying the abundance of ST3Gal1 protein levels in response to androgens. Whilst we appreciate that western blotting can be useful for interrogating changes in proteins such as post-translational modifications, we opted to use the more sensitive and quantitative ELISA technique as the most appropriate method to address this question.*

To help with abbreviations, the figure legend text has been changed to read “Protein level quantification of ST3Gal1 expression in LNCaP cells cultured with or without 10 nm R1881 synthetic androgens (A+) for 24 hours. Protein quantified using a pre-validated ST3Gal1 sandwich ELISA.”

Reply: *This is a good point from the reviewer, and we have changed the axis title in figure 1E to read “Log2 fold change of ST3GAL1 mRNA”.*

5. Authors suddenly talk about the AR variants without give the context of why authors felt a need of addressing the need of the question AR variants. As reader, I feel that it doesn't add anything substantial to the story. May be AR variant s figure can be added in the supplementary rather in the main figure.

Reply: *At the beginning of this study, we set out to comprehensively investigate how AR signalling alters ST3Gal1 levels in prostate cancer. Given recent literature referenced on line 122 demonstrating the clinical importance of AR variants in the progression and treatment resistance of advanced prostate cancer we feel that the findings are of interest and are an important finding in the study.*

6. The section, 159-163 - the overall sialylation is reduced, was there any placebo taken for the study?

Reply: *We only detected a change in α 2-3-sialylation, not total sialylation. Knock-out cells were compared with TRAMP-C2 cells which were edited using a non-targeting version of the PX459 CRISPR vector (no guide RNA). These non-targeting cells were selected using Puromycin and single-cell cloned using the methods used to generate the ST3Gal1 knockout. To reflect this, we have changed the text to read (line 175): “We confirmed successful gene knockout of St3gal1 in TRAMP-C2 and a subsequent reduction in α 2-3-sialylation compared with TRAMP-C2 cells transfected with a non-targeting CRISPR vector”.*

7. Figure 2B Proteins expression levels using should be included in the main figure. The supplementary blots don't meet the standard quality for inferring the result.

Reply: *As mentioned previously, we confirmed an increase in ST3GAL1 mRNA and protein using RT-qPCR and ELISA. Our focus was quantifying the abundance of ST3Gal1 protein levels in response to enzalutamide. We opted to use the more sensitive and quantitative ELISA technique as the most appropriate method to address this question. There are no supplementary western blots to accompany figure 2B.*

8. Please give bit more detail of the number or ratio when referring slight increase in the spheroid's experiments – line 171 figure 3D.

Reply: *This is an excellent point. We have now quantified the diameter of the spheroids formed for this experiment. We found that there was a 26% increase in the diameter of the spheroids formed by St3gal1^{-/-} TRAMP-C2 cells but this difference was not significant when a t-test was performed (P=0.08). We have now added this information to the text and have added the quantification data to supplementary figure 3F).*

9. Authors refer to the proliferation assay in figure 3G-H, WST assay is based on metabolic based assay, when addressing any metabolic questions its ideal to check proliferation by cell counts or DNA content. Authors may remove the proliferation figure or rather perform the assay as per the suggestion.

Reply: *We have taken on board the reviewers important point about the WST-1 assay. We have kept the WST-1 data in as it is important for addressing an point raised by reviewer 2 (addressed below). However, we have repeated the proliferation experiments for all of the cell lines mentioned using a sulforhodamine B (SRB) colourmetric assay. The dye binds to amino acids in basic cellular proteins and can be used to quantify total protein mass which can be related to cell number. These data have been added to supplemental figures 3E, 3O, 3P, 3V and 3W. We found no differences in protein content between any of the comparisons. This is in support of our findings using colony forming assays to suggest that ST3Gal1 does not affect proliferation in prostate cancer. Changes to the manuscript text are highlighted using tracked changes.*

10. Figure 4 authors starts to prolife siglec 9 and 7 but in the figure 4H only siglec 9 has been shown. Immunohistochemistry of siglec 7 is missing. Authors needs to comment why it has not it has not been considered.

Reply: *We agree that the reason for this is not clear in the manuscript. We have tried multiple siglec-7 antibodies from various vendors When we used these to perform immunohistochemistry on human prostate tissue we weren't confident that the staining patterns that we observed were genuine or specific. We have amended the manuscript text through tracked changes to reflect this. This has been added to line 238 and reads "We attempted to quantify the number of Siglec-7⁺ cells in this cohort of patients however attempts to optimise Siglec-7 antibodies for immunohistochemistry on prostate tissue were unsuccessful."*

11. Figure 4D, In the legends section please do include the full form of abbreviation for AMACR so reader doesn't have to investigate the text.

Reply: *This is a good spot and we have now added the full protein name has been added to the figure legend.*

12. Line 220, please elaborate bit more in detail. Line 222: what is the standard of care hormone therapies are? Please elaborate more in detail.

Reply: *For clarity we have now changed the text to read "standard of care anti-androgen therapies".*

13. Figure 5 B please include ruler when showing the immunofluorescence/ confocal pictures.

Reply: *This is an important point. Scale bars have now been added to all images throughout the manuscript and defined in figure legends.*

14. Paragraph – 269 -277 can be simplified for clear readability.

Reply: *We have tried to add clarity and aid readability to this section of the manuscript. Changes to the text are marked using tracked changes and the section now reads, "As observed previously, *St3gal1*^{-/-} cells implanted in IgG control mice failed to engraft (**Figure 5K**). Strikingly, depletion of CD8⁺ T cells resulted in a 75% engraftment rate of *St3gal1*^{-/-} cells, suggesting that the failure to engraft was, in part, due to CD8⁺ T cell dependant mechanisms (**Figure 5L**). Analysis of tumour growth kinetics showed a delay in tumour growth in anti-CSFR1 treated mice compared with anti-CD8 treated animals (**Figure 5M**). However, macrophage depletion also resulted in a 75% engraftment rate, demonstrating a key role for macrophages in mediating *St3gal1* driven immune suppression. Given that macrophages are not conventionally considered to have direct cytotoxic capabilities, we hypothesise that following depletion of *St3gal1* and subsequently Siglec-E ligands, macrophages may be re-educated towards an anti-tumour phenotype which could have secondary effects on cytotoxic effector cells such as CD8⁺ T cells."*

15. Discussion: line authors mention about the complexity of the glycol immune axis. How do they plan to address cell specific consequences? Authors shall reflect in detail, how this study is adding to already existing knowledge in respect to sialylation or ST3 Gal1 impact in cancer? authors discuss about the complexity line 347 – 349 does not give clear information nor about the current therapeutic therapies neither the unintended consequences.

Reply: *We have rewritten this section to aid with clarity.*

16. Discussion could be written better in way emphasizing the importance of the study, what it is contributing to existing research and drawbacks clearly. Addressing question in glycobiology is complex and when trying to establish the correlation with cancer is more complicated. Authors can discuss little bit about the specificity where they suggest that glycol-immune check points can be exploited for the new therapies.

Reply: *We have added text to this section to add clarity around the exciting glycan targeting therapies available.*

Reviewer #2 (Remarks to the Author):

The present manuscript investigates the role of glyco-immune checkpoints in the progression of advanced prostate cancer (PC) and its resistance to immunotherapy. It draws attention to the unmet therapeutic need for castrate-resistant prostate cancer, with novel targets being sought after for better treatment.

The authors explore the potential of the glyco-immune checkpoint, specifically the sialyltransferase ST3Gal1 and its associated α -2-3-linked sialylation patterns, as a novel target for combination immunotherapy. This builds on the premise that prostate tumors have the necessary anti-tumor effectors for an immunotherapy response but are locked by immune suppression, a majority of which is maintained by AR-independent mechanisms.

The study provides compelling evidence of AR-dependent glycosylation changes in PC. The data supporting an inverse correlation between AR signaling and ST3Gal1 expression in prostate tumors is strong. The authors convincingly show that enzalutamide, an anti-androgen therapy, increases levels of ST3Gal1-driven α -2-3-sialylation and synthesizes immunosuppressive Siglec-7 and Siglec-9 ligands in PC. They propose that enzalutamide may inadvertently upregulate these suppressive glyco-immune checkpoints, thereby creating a compelling rationale for considering Siglec targeting therapies in combination with enzalutamide.

The authors demonstrate the implications of ST3Gal1-sialoglycan-Siglec axis in anti-tumor biology. It shows that depleting ST3Gal1 associated sialoglycans or targeting their respective Sigecls may boost immune tumor clearance. However, the specific mechanisms underlying the correlation between AR signaling and ST3Gal1 expression remain unclear and warrant further investigation.

Moreover, the paper highlights the urgent need to better understand the cell-type specific

consequences of systemic therapies for optimizing treatment plans. The suggestion of combining therapies targeting glyco-immune checkpoints with current standard of care treatments is relevant and timely given the ongoing development of such therapies.

While a few additional experiments could further strengthen the manuscript, they are not required and should not delay the publication of the manuscript in its current form in any way.

Reply: *We thank the reviewer for their kind and positive feedback regarding the importance of the study. We have addressed individual points below:*

1) Seeing the induction of ST3Gal1 expression by enzalutamide-treatment, it would be interesting to compare the efficacy of enzalutamide-treatment on wt and ST3Gal1 knockout/knockdown cells.

Reply: *This is an excellent suggestion from the reviewer and something that we had not considered. We performed additional experiments to address this question. We treated LNCaP cells (empty vector control and ST3Gal1 knockdown) with a range of enzalutamide concentrations (1nM to 30 μ M) and measured cell viability after 72 hours using an SRB assay. We found that loss of ST3Gal1 had no effect on LNCaP response to enzalutamide when compared with empty vector control cells. We have included this data in supplementary figure 3X and have made reference to it in the text through tracked changes.*

2) Similarly, does the upregulation of ST3Gal1 affect the response to immunotherapy in combination with enzalutamide?

Reply: *We thank the reviewer for this comment and agree that this is the natural progression of the work presented here and is of great importance. There is clear rationale to study the role of ST3Gal1, and the immunoreceptors Siglec-7 and Siglec-9 in mediating response to immunotherapy. Studies on other cancer types have demonstrated that targeting the sialoglycan-Siglec axis does sensitise solid tumours to immune checkpoint blockade, specifically anti-PD-1 and anti-CTLA-4. This is an ongoing project in our lab and is beyond the scope of the study presented here. Understanding how other immune checkpoints are expressed and co-expressed with Siglecs in prostate cancer is important for designing the most promising therapeutic strategies. Our preliminary data demonstrates that SIGLEC7 and SIGLEC9 are positively correlated with levels of PD-L1 (CD274) (the ligand for the immune checkpoint PD-1). Conversely, they are negatively correlated with levels of the B7-H3 (CD276). B7-H3 is a promising new target for immune checkpoint blockade, specifically for those patients who express low levels of PD-L1. This highlights the potential for Siglec targeting therapies to benefit a range of immunotherapy resistant patients but also the breadth of work required to develop the most effective combination therapy.*

To emphasise their clinical importance, we have included the analysis of SIGLEC7/SIGLEC9 co-expression with immune checkpoints (Figure 5F) and have referenced this in the text through tracked changes.

3) Most pathways identified in Fig. 1B are linked to metabolism. Follow-up of targets beside AR signaling might strengthen the overall role of ST3Gal1 in PRAD.

Reply: *We agree with reviewer two that this is an interesting finding. We have performed WST-1 assays on ST3Gal1 knockdown, knock-out and over expression cell lines (figure 3 and supplemental figure 3). This assay provides a read-out of metabolic activity in cells. We found no significant change in metabolic activity with ST3GAL1 gene modulation across any of our cell lines. This is a crude way to study metabolism and our findings in figure 1B indicate that this could be studied further. Given the specialised and complex experiments and expertise needed to correctly interrogate changes in metabolism it is currently beyond the scope of this study but something that we will reach out to collaborators for to investigate further.*

4) A single cell/flow cytometric analysis of the expression pattern of Siglec-7/9 receptors on immune cells in human PC samples would add an additional dimension to support the immunofluorescence staining's.

Reply: *We agree with the reviewer that a single-cell analysis of the receptors in human prostate tissue is needed to corroborate our findings. We have reached out to our collaborators who recently performed a single-cell transcriptomic analysis of human prostate tissue (Joseph et al. 2021, now added to the reference list). They looked at transcript expression of SIGLEC7 and SIGLEC9 across cell-types and confirmed that they are predominantly expressed on myeloid cells, including monocytes and dendritic cells. We have now added this data to supplemental figure 5A and 5B and have referenced this in text through tracked changed.*

5) Similarly, a single-cell analysis of cells expressing Siglec-7/9 ligands might provide valuable insights.

Reply: *This is another excellent point raised by the reviewer and something which we are planning to do in our ongoing work. We plan to do this by flow cytometry using fresh tissue from prostate cancer patients. Currently we do not have the ethics in place to collect the necessary tissue but expect this to be in place later this year. This will be included in further studies in this area.*

Overall, the manuscript makes significant contributions to the understanding of glyco-immune checkpoints in prostate cancer and their potential as therapeutic targets. Future work should

focus on elucidating the precise mechanisms underlying these observations and conducting clinical trials to test the proposed combination therapies.

Reply: *We thank the reviewer for important points raised which have strengthened the manuscript.*

Reviewer #3 (Remarks to the Author):

In this manuscript Garnham et al. examine an intriguing inverse correlation between expression levels of androgen signaling pathway members and the sialyltransferase ST3Gal1. The authors convincingly demonstrate that androgen receptor antagonists currently used in the clinic increase both the expression of the ST3Gal1 enzyme and the abundance of its' products, 2,3-sialylic acid structures, on the cell surface. Prior reports have shown that these sialic acid structures can act as Siglec ligands, and the authors expand this knowledge by showing that 2-3-sialylated patterns are Siglec-7/-9 ligands on prostate tumors. Finally, they provide some evidence that these ST3Gal1 upregulated Siglec ligands are benefiting cancer growth through immune inhibitory macrophages.

This manuscript has fascinating implications for the future of prostate cancer therapy. In particular the finding that treatment of prostate tumors with an androgen receptor antagonist currently used in the clinic leads to the upregulation of immune inhibitory Siglec ligands and a more inhibited TIME may help explain why a significant percentage of patients are not responsive to conventional immune therapies. While the exact mechanism of how androgen receptor signaling is linked to ST3Gal1 expression is not elucidated in this work, the findings are novel and interesting to the field of biology. The data is well-presented, and a major strength that the authors make the effort to repeat their findings in multiple of cell lines, spheroid models, animal models, as well as using human data and tissue biopsies throughout the manuscript.

Reply: *We thank the reviewer for their kind comments about the novelty and quality of the manuscript. We have addressed individual points below.*

I have two requests for further information or possible experiments:
1. In lines 235-240 / Fig. 5B & C. It is mentioned that both SIGLEC7 and SIGLEC9 cluster with an anti-inflammatory macrophage gene signature. The manuscript then confirms Siglec-9 expression colocalization on CD14+ and CD163+ myeloid cells in patient biopsies, but no mention is made of having looked for Siglec-7 expression there. Since the bulk of the paper focuses on both of these receptors it would be ideal to include these results either in the main

figure or the supplementary Figs. Alternatively, an explanation of why the authors elected to proceed with only Siglec-9 colocalization could be provided.

Reply: *We agree that the reason for this is not clear in the manuscript. We have tried multiple siglec-7 antibodies from various vendors. When we have used these to perform immunofluorescence on human prostate biopsies we were not confident that the staining patterns observed were genuine or specific. We have amended the manuscript text through tracked changes to reflect this. We have added text to line 262 which reads, “Due to a lack of specific Siglec-7 antibodies that we were confident about, we could not perform co-immunofluorescence experiments for Siglec-7 and myeloid markers.” This is also marked in the text using tracked changes.*

2. Similarly, in Figure 4. The bulk of the figure focuses on looking at Siglec-7/-9 ligands on multiple cell lines and in patient biopsies, but for the tissue microarray quantification (Fig. 4H & I) only Siglec-9 is investigated. Ideally, there would also be data for Siglec-7 here or in supplementary data, even if the data are negative, but there may be reasons why this could not be/was not done.

Reply: *Similarly, we spent time optimising the immunohistochemistry staining protocol for Siglec-7 ligands. We tried changing the antigen retrieval, tried a range of concentrations and adding additional blocking steps. Ultimately, we were unhappy with the quality and specificity of the staining and weren't satisfied that we would be able to quantify the results. We have added the following text to our manuscript “We failed to successfully optimise a staining protocol using Siglec-7-FC reagents to detect Siglec-7 ligands in patient tissue. For this reason, it was excluded from our study” and have marked the changes using tracked changes.*

3. In the Figure 1D legend – state how the protein was quantified

Reply: *The following was added to the figure legend: “Protein quantified using a pre-validated ST3Gal1 sandwich ELISA”.*

4. In Figure 1I legend – could you state how many were in each group (amplified) and (unaltered).

Reply: *This is an important point. N numbers have been added to the figure legend for each group.*

5. In Figures 1A , 2I, 3D, 4D, 4E, 4H, 4I, 5B, 5C, there is either no scale bar, or the scale bar is present but not labelled with a number on the figure or in the legend.

Reply: *This is a good point and scale bars have been added to all images and defined in figure legends.*

6. For Fig. 2H, this is likely a glitch with this particular upload – but the background for this subfigure is neon yellow in the version uploaded and certain cell populations can't be seen.

Reply: *This is a glitch with the upload and we have tried to correct this in the resubmission.*

7. Figure 2H (right half): Coloring scheme is undifferentiable to red/green colorblind folks, would be preferable to change this.

Reply: *We were unsure which figure the reviewer is referring to and as a result have not made any subsequent changes.*

8. Supplementary Figure 1D has a textbox pasted over the figure.

Reply: *Thank you for pointing this out, this has now been changed in the figures.*

9. There are some single representative tissue samples/TMA/confocal images in the paper, can some other images collected for these experiments be added to the supplementary figures or to a database?

Reply: *This is a good point raised by the reviewer. We have now added additional images of immunohistochemistry and immunofluorescence experiments to the corresponding supplemental figure for each of the representative images. These can be found in supplemental figures 1A, 2C, 4D, 4E, 4F and 5A.*

10. The manuscript occasionally mentions a representative image in a figure – each time a representative image is used please include how many replicates/images were taken for each experiment (for example, how many spheroids were imaged for each group in Fig. 3D?)

Reply: *This is an important point raised by the reviewer. We have now added the requested information to the figure legends.*

11. In Figure 3D is there some way to quantify the spheroid sizes in your images?

Reply: *We have now quantified the diameter of the spheroids formed for this experiment. We found that there was a 26% increase in the diameter of the spheroids formed by *St3gal1*^{-/-} TRAMP-C2 cells but this difference was not significant when a t-test was performed ($P=0.08$). We have now added this information to the text and have added the quantification data to supplemental figure 3F).*

12. In all histogram-containing figures, I want to verify that the x-axis spacing is correct. If so, what scaling is this? It looks neither log scale nor bi-exponential - it might be good to add the scale information into the methods section or figure legends.

Reply: *We thank the reviewer for spotting this and making a very important point. We used the Red Matter App software to generate plots using a bi-exponential scale and this has been distorted during figure preparation. We have now corrected the axis label and also added the relevant text to the methods section stating 'Data was analysed and histograms generated using Red Matter App (v2.01) and plotted using a bi-exponential scale'.*

REVIEWERS' COMMENTS:

Please note that reviewer #1 has reassessed the manuscript and has expressed their satisfaction with the authors' responses to the reviewers' concerns, but has not left any specific comments to the authors.

Reviewer #2 (Remarks to the Author):

Thank you for your comprehensive and thoughtful responses to the points raised during the review. The revised manuscript successfully addresses all my remaining questions and should be published.

Response to reviewers' comments

We thank all of the reviewers for their time and a pleasant and helpful revision process. Their input has greatly improved our manuscript.

With regards to the point raised by reviewer 3 on x-axis labelling 'With regards to point 12 from reviewer #3, we note that you state that the spacing of the x-axis labels has been corrected for histogram-containing figures and mention in the Methods section that a bi-exponential scale has been used for the plots. However, it appears that the scaling for the x-axis of the histograms appears to be simple log scaling. We would be grateful if you could clarify this potential discrepancy between what is stated in the Methods section and the x-axes of the histograms.'

The reviewer is correct and this is a mistake on our part. We have corrected the methods section to state that the x-axis is a log scale.